# Confounding Robust Deep Reinforcement Learning: A Causal Approach

**Mingxuan Li**[1*]    **Junzhe Zhang**[2*]    **Elias Bareinboim**[1]
[1] Columbia University, [2] Syracuse University
[1]{ml,eb}@cs.columbia.edu, [2]jzhan403@syr.edu

## Abstract

A key task in Artificial Intelligence is learning effective policies for controlling agents in unknown environments to optimize performance measures. Off-policy learning methods, like Q-learning, allow learners to make optimal decisions based on past experiences. This paper studies off-policy learning from biased data in complex and high-dimensional domains where *unobserved confounding* cannot be ruled out a priori. Building on the well-celebrated Deep Q-Network (DQN), we propose a novel deep reinforcement learning algorithm robust to confounding biases in observed data. Specifically, our algorithm attempts to find a safe policy for the worst-case environment compatible with the observations. We apply our method to twelve confounded Atari games, and find that it consistently dominates the standard DQN in all games where the observed input to the behavioral and target policies mismatch and unobserved confounders exist.

## 1 Introduction

Over the last decade, reinforcement learning (RL) has gained significant popularity for solving complex sequential decision-making problems, primarily due to its integration with deep learning techniques [24, 52, 83]. This approach, known as deep reinforcement learning (deep RL), is particularly effective in high-dimensional state spaces [40, 61, 70, 84, 85, 88]. Deep RL addresses the challenges faced by earlier RL methods by extracting various abstractions from data in complex domains with minimal prior knowledge. For example, a classic algorithm known as Deep Q-Network (DQN) algorithm can efficiently learn from visual inputs containing thousands of pixels [61], enabling problem-solving capabilities comparable to humans in certain high-dimensional environments, such as Atari games for the first time. These achievements were followed by notable advancements in deep RL, including mastering the game of Go [89], defeating world-class professionals at the game of poker [64]. Deep RL has also shown potentials in various real-world applications like robotics [70, 88], autonomous driving [49] and protein design [40]. These advancements eventually culminated in Sutton and Barto receiving the Turing Award in 2025 [22].

This paper attempts to leverage the capabilities and insights of DQN, while identifying an important assumption embedded in this algorithm and its variants that does not necessarily hold in the real world. Particularly, we notice that it is often implicitly assumed through the (PO)MDPs [93] framework or explicitly enforced during the data-collection that no unmeasured confounder (NUC, [11, 78]) affects the observed action and the subsequent outcomes. When the NUC does not hold, the effect of the target policy is generally not *identifiable*, i.e., the model assumptions are insufficient to uniquely determine the value function from the offline data [72, 111]. On the other hand, partial identification is a line of methodologies that enable the derivation of informative bounds on target effects from confounded observations in non-identifiable settings [59]. It has been studied under the rubrics of causal inference [7, 115], econometrics [18, 35, 62, 75, 79, 92, 99], and dynamical systems [6, 15, 20, 63, 69]. More recently, researchers have been using partial identification methods to obtain

reliable off-policy evaluation in reinforcement learning [17, 39, 43, 44, 47, 48, 54, 67, 111, 113]. Despite these achievements, significant challenges still exist in applying partial identification for policy learning in complex and high-dimensional domains, including images and videos. We refer readers to App. A for a more detailed survey on partial identification and deep reinforcement learning.

This paper aims to address these challenges by investigating deep reinforcement learning algorithms from offline data over complex and high-dimensional domains, where the presence of unmeasured confounders could not be assumed away *a priori*. More specifically, our contributions are summarized as follows. (1) We introduce a novel DQN algorithm, which we call Causal DQN, capable of learning robust abstractions from confounded data over complex and high-dimensional domains with minimal prior knowledge. (2) We empirically demonstrate that our method significantly improves robustness and generalization under confounded observations and outperforms various DQN baselines across twelve popular Atari games. Due to space constraints, details of the experiment setup and additional experiments are provided in Apps. D and E. Videos of gameplay are included in the supplemental.

**Notations.** We will consistently use capital letters $(V)$ to denote random variables, lowercase letters $(v)$ for their values, and cursive $\mathcal{V}$ to denote the their domains. We use bold capital letters $(\boldsymbol{V})$ to denote a set of random variables and let $|\boldsymbol{V}|$ denote its cardinality of set $\boldsymbol{V}$. Finally, $\mathbf{1}_{\boldsymbol{Z}=\boldsymbol{z}}$ is an indicator function that returns 1 if event $\boldsymbol{Z} = \boldsymbol{z}$ holds true; otherwise, it returns 0.

## 2 Challenges Due to Unobserved Confounders

We will focus on a sequential decision-making problem in the Markov Decision Process (MDP, [77]) where the agent intervenes on a sequence of actions to optimize subsequent rewards. Standard MDP models focus on the perspective of learners who could actively intervene in the environment. Consequently, confounding is generally assumed away a priori. On the other hand, when considering off-policy data collected from passive observations, the learner does not necessarily have the liberty to control how the behavioral policy generates the data, giving rise to unobserved confounders in decision-making tasks [26, 43, 51, 81, 114]. In this paper, we will consider a generalized family of confounded MDPs [14, 44, 54, 112, 113] explicitly modeling the presence of unobserved confounders in the off-policy data generation.

**Definition 2.1.** A Confounded Markov Decision Process (CMDP) $\mathcal{M}$ is a tuple of $\langle \mathcal{S}, \mathcal{X}, \mathcal{Y}, \mathcal{U}, \mathcal{F}, P \rangle$ where (1) $\mathcal{S}, \mathcal{X}, \mathcal{Y}$ are, respectively, the space of observed states, actions, and rewards; (2) $\mathcal{U}$ is the space of unobserved exogenous noise; (3) $\mathcal{F}$ is a set consisting of the transition function $f_S : \mathcal{S} \times \mathcal{X} \times \mathcal{U} \mapsto \mathcal{S}$, behavioral policy $f_X : \mathcal{S} \times \mathcal{U} \mapsto \mathcal{X}$, and reward function $f_Y : \mathcal{S} \times \mathcal{X} \times \mathcal{U} \mapsto \mathcal{Y}$; (4) $P$ is an exogenous distribution over the domain $\mathcal{U}$.

Throughout this paper, we will consistently assume the action domain $\mathcal{X}$ to be discrete and finite, while the state domain $\mathcal{S}$ could be complex and continuous; the reward domain $\mathcal{Y}$ is bounded in a real interval $[a, b] \subset \mathbb{R}$. Consider a demonstrator agent interacting with a CMDP $\mathcal{M}$, generating the off-policy data. For every time step $t = 1, \ldots, T$, the environment first draws an exogenous noise $U_t$ from the distribution $P(\mathcal{U})$; the demonstrator then performs an action $X_t \leftarrow f_X(S_t, U_t)$, receives a

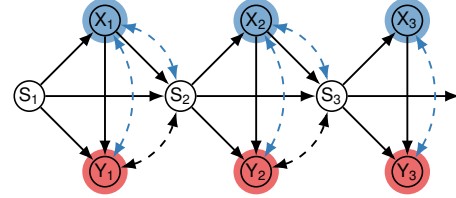

Figure 1: Causal diagram representing the data-generating mechanisms in a Confounded Markov Decision Process.

subsequent reward $Y_t \leftarrow r_t(S_t, X_t, U_t)$, and moves to the next state $S_{t+1} \leftarrow f_S(S_t, X_t, U_t)$. The observed trajectories of the demonstrator (from the learner's perspective) are summarized as the observational distribution $P(\bar{\boldsymbol{X}}_{1:T}, \bar{\boldsymbol{S}}_{1:T}, \bar{\boldsymbol{Y}}_{1:T})$, i.e.,

$$P(\bar{\boldsymbol{x}}_{1:T}, \bar{\boldsymbol{s}}_{1:T}, \bar{\boldsymbol{y}}_{1:T}) = P(s_1) \prod_{t=1}^{T} \left( \int_{\mathcal{U}} \mathbf{1}_{s_{t+1}=f_S(s_t, x_t, u_t)} \mathbf{1}_{x_t=f_X(s_t, u_t)} \mathbf{1}_{y_h=f_Y(s_t, x_t, u_t)} P(u_t) \right)$$

Fig. 1 shows the causal diagram $\mathcal{G}$ [10] describing the generative process of the off-policy data in CMDPs. More specifically, solid nodes represent observed variables $X_t, S_t, Y_t$, and arrows represent the functional relationships $f_X, f_S, f_Y$ among them. By convention, exogenous variables $U_t$ are often not explicitly shown in the graph; bi-directed arrows $X_t \longleftrightarrow Y_t$ and $X_t \longleftrightarrow S_{t+1}$ indicate the presence of an unobserved confounder (UC) $U_t$ affecting the action, state, and reward simultaneously. These bi-directed arrows (highlighted in blue) represent the unobserved confounders among action

$X_t$, reward $Y_t$, and state $S_{t+1}$ in the off-policy data, violating the condition of NUC [11, 78]. Such violations could lead to challenges in off-policy learning.

**Off-Policy Learning.** A policy $\pi$ in a CMDP $\mathcal{M}$ is a decision rule $\pi(x_t \mid s_t)$ mapping from state to a distribution over action domain $\mathcal{X}$. An intervention $\mathrm{do}(\pi)$ is an operation that replaces the behavioral policy $f_X$ in CMDP $\mathcal{M}$ with the policy $\pi$. Let $\mathcal{M}_\pi$ be the submodel induced by intervention $\mathrm{do}(\pi)$. The interventional distribution $P_\pi(\bar{\boldsymbol{X}}_{1:T}, \bar{\boldsymbol{S}}_{1:T}, \bar{\boldsymbol{Y}}_{1:T})$ is defined as the joint distribution over observed variables in $\mathcal{M}_\pi$, i.e.,

$$P_\pi(\bar{\boldsymbol{x}}_{1:T}, \bar{\boldsymbol{s}}_{1:T}, \bar{\boldsymbol{y}}_{1:T}) = P(s_1) \prod_{t=1}^{T} \left( \pi(x_t \mid s_t) \mathcal{T}(s_t, x_t, s_{t+1}) \mathcal{R}(s_t, x_t, y_t) \right) \tag{1}$$

where the transition distribution $\mathcal{T}$ and the reward distribution $\mathcal{R}$ are given by, for $h = 1, \ldots, H$,

$$\mathcal{T}(s_t, x_t, s_{t+1}) = \int_{\mathcal{U}} \mathbf{1}_{s_{t+1}=f_S(s_t, x_t, u_t)} P(u_t), \quad \mathcal{R}(s_t, x_t, y_t) = \int_{\mathcal{U}} \mathbf{1}_{y_t=f_Y(s_t, x_t, u_t)} P(u_t) \tag{2}$$

For convenience, we write the reward function $\mathcal{R}(s, x)$ as the expected value $\sum_y y \mathcal{R}(s, x, y)$. Fix a discounted factor $\gamma \in [0, 1]$. A common objective for an agent is to optimize its cumulative return $R_t = \sum_{i=0}^{\infty} \gamma^i Y_{t+i}$. We define the optimal action-value function $Q_*(s, x)$ as the maximum expected return obtainable by following any policy $\pi$, after seeing a state $s$ and taking an action $x$, $Q_*(s, x) = \max_\pi \mathbb{E}_{X_t \leftarrow x, \pi} [R_t \mid S_t = s]$. One could solve for an optimal policy by iteratively evaluating the action-value function using the *Bellman Optimality Equation* [12] given by,

$$Q_*(s, x) = \mathbb{E}_{X_t \leftarrow x} \left[ Y_t + \gamma \max_{x'} Q_*(S_{t+1}, x') \mid S_t = s \right] \tag{3}$$

In off-policy evaluation, the agent (i.e., learner) attempts to learn an optimal policy by leveraging the observed data generated by a different behavior policy $f_X$ (demonstrator). When there is no unmeasured confounder (NUC) introducing spurious correlations between action and subsequent outcomes, one could identify the parameterizations of the transition distribution $\mathcal{T}$ and reward function $\mathcal{R}$ from the observed data, i.e.,

$$\mathcal{T}(s_t, x_t, s_{t+1}) = P(s_{t+1} \mid s_t, x_t), \qquad \mathcal{R}(s_t, x_t, y_t) = P(y_t \mid s_t, x_t) \tag{4}$$

When the above identification formula hold, several off-policy algorithms have been proposed to estimate the effect of candidate policies from finite observations [38, 41, 42, 65, 76, 94, 104, 105]. Together with the computational framework of deep learning, these methods could be further extended to complex domains [61, 70, 84, 85, 88]. However, NUC could be fragile in practice and does not necessarily hold due to some violations in the generative process. In these situations, applying standard off-policy methods may fail to converge to an optimal policy, despite using powerful deep learning models. The following example illustrates such challenges in a classic Atari game.

**Example 1** (Confounded Pong). Consider the Pong game in the classic Atari suite. As for the behavior policy, we use a pre-trained high-performing actor-critic agent [2] with residual blocks [29] and Long Short-Term Memory (LSTM, [32]) layers. Fig. 2a shows a saliency map visualizing the learned policy. Simulation results show that this policy is optimal. For example, this agent can deliver a "kill shot" by directing the ball to a location that the opponent is unlikely to intercept given its current location. We use this agent as the demonstrator in the off-policy learning task.

We now consider an alternative agent that learns to play Pong by observing the demonstrator's gameplay trajectories. This learning agent has a simpler neural network architecture and an impaired sensory capability: it can only observe movements in its nearby surroundings. Fig. 2b shows the learner's visual input; the board's left-hand side and the upper side, including the opponent's position and score, is now masked. In this case, the opponent's position becomes an unobserved confounder, introducing spurious correlations between the demonstrator's action and observed outcome. For example, the behavioral policy tends to hit the ball toward the center only when the opponent is positioned at either corner and unable to return it. As a result, center shots appear more effective than they truly are, due to confounding resulted from the masked opponent's position.

To validate whether the standard deep RL algorithms are robust to confounding biases, we train two DQN learners on masked trajectories from the demonstrator: one is the standard convolutional neural network based DQN (Nature DQN, [61]), the second one is an LSTM-based [28]. We also include a

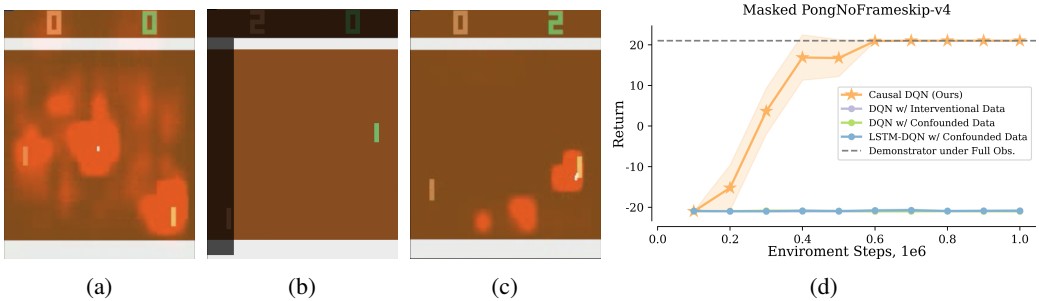

(a)       (b)       (c)       (d)

Figure 2: (a) A saliency map of the behavioral policy in Pong that tracks the opponent's location and score board; (b) a confounded Pong game where the opponent's location and score board is masked; (c) a saliency map of the conservative policy focusing on only itself and the ball; (d) the average return of our causal DQN and the standard DQN baselines. Baseline curves are overlapped.

standard DQN that learns directly in the masked Pong game without confounded demonstrations as a baseline. The simulation results, shown in Fig. 2d, indicate that none of those DQN variants is able to converge to an effective policy. Learning directly with impaired observation in Atari is challenging, while incorporating confounded demonstrations directly also does not enhance the convergence of DQN agents; instead, it negatively impacts their learning performance. ∎

## 3 Confounding Robust Deep Q-Learning

In this section, we will introduce partial identification methods for off-policy learning that are robust to unobserved confounding. Recently, Zhang & Bareinboim [113] extended the well-celebrated Bellman equation to allow one to lower bound the state-action value function $Q_\pi(s, x)$ with a closed-form solution $\underline{Q}_\pi(s, x)$, which can be consistently estimated from the confounded observations. We extend this result to obtain a lower bound for the optimal value function.

**Proposition 3.1** (Causal Bellman Optimality Equation). *For a CMDP environment $\mathcal{M}$ with reward signals $Y_t \in [a, b] \subseteq \mathbb{R}$, its optimal state-action value function $Q_*(s, x) \geq \underline{Q}_*(s, x)$ for any state-action pair $(s, x) \in \mathcal{S} \times \mathcal{X}$, where the lower bound $\underline{Q}_*(s, x)$ is given by as follows,*

$$\underline{Q}_*(s, x) = P(x \mid s)\left( \widetilde{\mathcal{R}}(s, x) + \gamma \sum_{s', x'} \widetilde{\mathcal{T}}(s, x, s') \max_{x'} \underline{Q}_*(s', x') \right) \tag{5}$$

$$+ P(\neg x \mid s)\left( a + \gamma \min_{s'} \max_{x'} \underline{Q}_*(s', x') \right) \tag{6}$$

*where $P(x \mid s) = P(X_t = x \mid S_t = s)$ and $P(\neg x \mid s) = 1 - P(x \mid s)$; $\widetilde{\mathcal{T}}$ and $\widetilde{\mathcal{R}}$ are nominal transition distribution and reward function computed from the observational distribution, i.e.,*

$$\widetilde{\mathcal{T}}(s, x, s') = P(S_{t+1} = s' \mid S_t = s, X_t = x), \qquad \widetilde{\mathcal{R}}(s, x) = \mathbb{E}[Y_t \mid S_t = s, X_t = x] \tag{7}$$

Prop. 3.1 lower bounds the expected return of an optimal policy $\pi^*$ using the return of a pessimistic policy $\underline{\pi}^*$ that optimizes a worst-case CMDP instance $\underline{\mathcal{M}}$ compatible with the observational data. The lower bound is set as the expected return of the pessimistic policy $\underline{\pi}^*$ in the worst-case CMDP $\underline{\mathcal{M}}$, i.e., $\underline{Q}_*(s, x) \triangleq Q_{\underline{\pi}^*}(s, x; \underline{\mathcal{M}})$. Since $\pi^*$ is optimal in the ground-truth CMDP environment $\mathcal{M}$, we must have $Q_*(s, x; \mathcal{M}) \geq Q_{\underline{\pi}^*}(s, x; \mathcal{M}) \geq Q_{\underline{\pi}^*}(s, x; \underline{\mathcal{M}})$. Optimizing the lower bound in Prop. 3.1 leads to a pessimistic policy with a performance guarantee in the ground-truth environment. Among quantities in the lower bound Prop. 3.1, nominal transition distribution $\widetilde{\mathcal{T}}$ and nominal reward function $\widetilde{\mathcal{R}}$ are functions of the observational distribution, and, at least in principle, are consistently estimable from the sampling process. The lower bound $\underline{Q}_*(s, x)$ can thus be further written as:

$$\underline{Q}_*(s, x) = \mathbb{E}\left[ \mathbf{1}_{X_t = x}\left( Y_t + \max_{x'} \underline{Q}_*(S_{t+1}, x') \right) + \mathbf{1}_{X_t \neq x}\left( a + \min_{s'} \max_{x'} \underline{Q}_*(s', x') \right) \mid S_t = s \right] \tag{8}$$

---

**Algorithm 1** Causal Deep Q-Learning (`Causal-DQN`)

---

1: Initialize replay memory $\mathcal{D}$
2: Initialize action-value function $\underline{Q}_*(\cdot; \theta)$ with random weights $\theta$
3: **for** episodes $= 1, \ldots, M$ **do**
4:     Sample initial state $s_1$
5:     **for** $t = 1, \ldots, T$ **do**
6:         Observe an action $x_t$ taken by the demonstrator and subsequent reward $y_t$ and state $s_{t+1}$
7:         Store transition $(s_t, x_t, y_t, s_{t+1})$ in $\mathcal{D}$
8:         Sample a minibatch of transitions $\{(s_i, x_i, y_i, s_{i+1})\}_{i=1}^B$ from $\mathcal{D}$
9:         Set value target $w_i(x)$ for every action $x \in \mathcal{X}$ w.r.t sample $(s_i, x_i, y_i, s_{i+1})$,

$$w_i(x) = \begin{cases} y_i + \gamma \max_{x'} \underline{Q}_*(s_{i+1}, x'; \theta) & \text{if } x = x_i \\ a + \gamma \min_{s'} \max_{x'} \underline{Q}_*(s', x'; \theta) & \text{if } x \neq x_i \end{cases} \tag{11}$$

10:         Perform a gradient descent step on $\sum_x \left(w_i(x) - \underline{Q}_*(s_i, x; \theta)\right)^2$ according to Eq. (10)
11:     **end for**
12: **end for**

---

In the above equation, $Y_t$ and $S_t$ are observed variables drawn from the nominal reward function $\widetilde{\mathcal{R}}$ and transition distribution $\widetilde{\mathcal{T}}$ in Eq. (7), respectively. Fig. 3 shows a backup diagram illustrating this update step. Like the standard Bellman optimality equation (Eq. (3)), Eq. (8) recursively updates the value function based on the current estimates of the optimal value function. On the other hand, Eq. (8) explicitly accounts for the off-poicy nature of the confounded observations: when the behavior policy takes the same action $x_t = x$ as the target action, the update follows standard Bellman equation and uses the next sampled state $s_t$; when the sampled action $x_t \neq x$ differs from the target, our algorithm updates, instead, using the value function associated with the next worst-case or best-case state $s^*$, corresponding to the estimation of the lower bound and upper bound respectively.

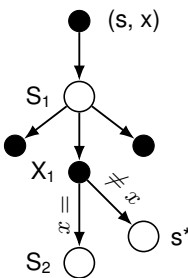

Figure 3: Backup diagram for causal deep Q-learning.

By using the causal Bellman equation of Eq. (8) as an iterative update, one could apply standard value iteration to obtain a robust policy against the confounding bias in the off-policy data [113]. In practice, however, this approach could be computationally challenging for complex and high-dimensional domains. Like many deep reinforcement learning algorithms [61], we will use a neural network with weights $\theta$, called Q-network, to approximate the lower bound over the state-action value function, i.e., $\underline{Q}_*(s, x; \theta) \approx \underline{Q}_*(s, x)$. We will train a Q-network by minimizing a sequence of loss functions $L_i(\theta_i)$ at each iteration $i$. Formally,

$$L_i(\theta_i) = \mathbb{E}_{s \sim \rho(\cdot)} \left[ \sum_x \left(W_i(x) - \underline{Q}_*(s, x; \theta_i)\right)^2 \right] \tag{9}$$

where function $W_i(x)$ is defined as the right-hand side of the update procedure Eq. (8); and $\rho(s)$ is the state occupancy distribution in the observed Markov chain under the behavioral policy. The parameters from the previous iteration $\theta_{i-1}$ are held fixed when optimizing the loss function. Note that the above loss function attempts to minimize the error of the Q-network bound over all actions. The reason is that, in the causal Bellman update (Eq. (8)), the next observed action contains information about the lower bound across all actions, regardless of whether it matches the actual action taken. Differentiating the loss function with respect to the weights, we arrive at the following gradient,

$$\nabla_{\theta_i} L_i(\theta_i) = \mathbb{E}_{s \sim \rho(\cdot)} \left[ \mathbb{E} \left[ \sum_x \left(W_t(x) - \underline{Q}_*(s, x; \theta_i)\right) \nabla_{\theta_i} \underline{Q}_*(s, x; \theta_i) \mid S_t = s \right] \right] \tag{10}$$

Details of our proposed algorithm, called Causal Deep Q-Learning (`Causal-DQN`), are provided in Algo. 1. Like the standard DQN [61], our algorithm utilizes experience replay [57]. Particularly, it stores trajectories observed at each time step, represented as $(s_t, x_t, y_t, s_{t+1})$, in a replay memory $\mathcal{D}$ that is pooled from many episodes. During the inner loop of the algorithm, we apply minibatch stochastic gradient descent to samples of experience $(s_i, x_i, y_i, s_{i+1}) \sim \mathcal{D}$, which are randomly

drawn from the pool of stored samples. On the other hand, our proposed causal algorithm made the following augmentations compared to the standard DQN. First, `Causal-DQN` is an off-policy learning algorithm and does not actively intervene in the environment. At step 6, instead of exploiting the Q-network being trained, it queries the demonstrator to generate a confounded transition sample. Second, at Step 9 during the experience replay, Causal-DQN utilizes the causal Q-learning updates of Eq. (10), which is robust to the potential presence of confounders. Particularly, when the observed action $x_i$ is equal to the evaluated action $x$, the algorithm follows the standard Q-learning update. Otherwise, it performs the update using a lower-bound $a$ over the immediate reward and the value function at the worst-case next state $s'$. The worst-case state $s'$ is empirically estimated by repeatedly sampling the next possible states at random, and taking the one with the smallest value function estimate. These augmentations improve Causal-DQN over its non-causal counterpart in terms of robustness and sample efficiency, as it is able to utilize the abundant observational data to improve the evaluation of the state-action value function.

**Example 2** (Confounded Pong continued). Consider again the confounded Pong game described in Example 1. We train a Causal DQN agent with the masked observed trajectories. Fig. 2c shows the saliency map visualizing the learned policy. Our proposed method learns a conservative policy focusing on only tracking the ball location instead of opponent's location. Simulation results show that this conservative policy is able to achieve comparable performance to the optimal demonstrator using the full board information. Analyzing the gameplay video reveals that our causal DQN agent learns to proactively place the ball in either corner, where the hard-coded AI opponent is unable to return. See the gameplay video in the supplementary materials for more details. ∎

## 4  Experiments

In this section, we aim to demonstrate the robustness and performance improvement of our proposed `Causal-DQN` under confounded settings. For a comprehensive evaluation of `Causal-DQN`, we choose twelve popular Atari games from the Gymnasium benchmark [100] and design the corresponding confounded versions. See below and also App. C for our detailed design of confounded Atari games. For a fair comparison, we also use vanilla DQN with little modifications as the baselines we test. More specifically, the baselines include (1) a CNN-based DQN with confounded demonstrator data (Conf. DQN), (2) an LSTM-based DQN with confounded demonstrator data (Conf. LSTM-DQN), and (3) a CNN-based DQN trained directly under masked observations without confounded data (Interv. DQN). For (1-2), the DQN agent will query the demonstrator for data samples as the `Causal-DQN` does. For (3), the DQN agent uses its own policy to sample environment transitions.

For each game, we train the agent for 1 million environment steps. We use 20 parallel environments to collect samples. At each parallel environment step, a minibatch is sampled to train the agents, equivalent to an update frequency of 20. We use a batch size of 512, a replay buffer of 100K in size, and a learning rate of $5e{-}4$ to accelerate convergence. Other hyperparameters are the same as in [61]. All results presented in this section are evaluation performances where we test each trained agent in the Atari game with masked observations. Curves in Fig. 8 are generated by evaluating the agent periodically in a separate evaluation environment, not from training returns.

**Data Preparation and Model Architecture.** We use the standard Atari game preprocessing for the input to the agents except that we resize the input to be 64×64 to align with the size requirement of the demonstrator [2], a competitive actor-critic agent with deep residual blocks and LSTM layers. Other differences in input preprocessing for the demonstrator are that (1) the demonstrator takes a single colored frame as the input per each time step, and (2) the demonstrator has access to the original full screen observation while the learners only has access to a masked partial screen. For all DQN tested, we adopt Double DQN [101] to stabilize learning. The CNN version follows the set up of Nature DQN [61] and the LSTM version only replaces the second-to-last linear layer with an LSTM cell. See App. D for detailed model architectures.

**Designing the Confounded Atari Games.** In the confounded Atari games, we mask out certain areas in each game's observation to prevent the agent from using spurious correlated features. To find such spurious visual artifacts used by the demonstrators, we apply a perturbation-based approach [25] to visualize saliency maps of both the actor and the critic of the demonstrator [2]. As shown in Fig. 2a, in the Pong game, the demonstrator is constantly checking the score board and also the opponent's paddle locations, neither of which is necessary for winning since the opponent in Pong has a fixed policy regardless of the current score and as long as the agent shoot back the ball, there is a

chance to score. Thus, an intuitive optimal policy should only look at the ball location and the agent's own paddle location to decide the move. In Fig. 2b, we mask out those areas and use the masked out observation as the state input ($s_t$) to DQNs while the masked area becomes the confounder $u_t$ which can only be observed by the demonstrator's policy, i.e., $x_t \leftarrow f_X(s_t, u_t)$.

For the remainder of this section, we will first present a few other notable confounded Atari games and discuss the performance of our proposed `Causal-DQN` in all 12 confounded Atari games. Specifically, despite confounding bias, our proposed causal agent is able to obtain an effective policy under masked observations from the demonstrator's trajectories. The learned policies demonstrate conservative behaviors aligned with human intuitions. Overall, `Causal-DQN` consistently dominates its non-causal vanilla counterparts in all 12 confounded Atari games in performance.

**Confounded Boxing.** In the original Boxing, the player controls the white agent to punch the black one to score. The party with the higher score when the time runs out, or any party hitting 100 first, wins the game. The demonstrator's policy picks up an aggressive "brawler" style which pressures the opponent and trades blows in the center of the arena. Fig. 4a shows the saliency map of the demonstrator's policy.

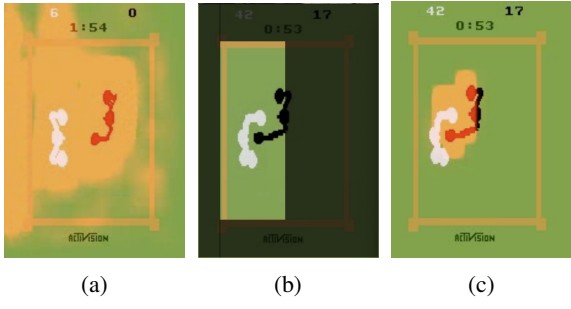

(a)                (b)                (c)

Figure 4: (a) A saliency map of the demonstrator's policy; (b) a confounded Boxing game where only the left half of the arena is visible; (c) a saliency map of `Causal-DQN`'s policy.

In the confounded Boxing, we mask out the score/remaining time, the outer area of the arena, and the right half of the arena. In words, the agent has impaired eyesight and cannot keep track of the current score and remaining time. Our `Causal-DQN` agent picks up a conservative "rope-the-dope" boxing style which focuses on defending its ground on the left-hand side of the arena. Fig. 4c shows the saliency map of such a conservative policy. Perhaps surprisingly, simulation results, shown in Table 1 and Fig. 8, reveal that despite the limited sensory capabilities, `Causal-DQN` agent can still achieve similar performance as the optimal demonstrator, defeating the hard-coded AI opponent.

**Confounded Gopher.** In the Gopher game, the player controls a farmer with a shovel, tasked with protecting a garden of carrots from a mischievous gopher. The gopher repeatedly attempts to tunnel underground to steal the carrots. The player must move horizontally across the screen to block the gopher's digging attempts by filling holes. A shortcut strategy is to follow the gopher's location underground closely so that the farmer is always close to those newly completed holes. Fig. 5a shows the saliency map of the demonstrator's policy. Our analysis reveals that it manages to pick up a "proactive" playing strategy, which actively tracks the gopher's location and uses this information to adjust the farmer's position.

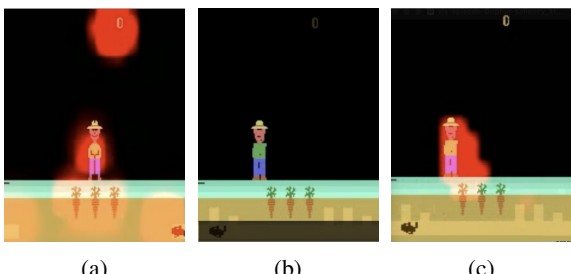

(a)                (b)                (c)

Figure 5: (a) A saliency map of the demonstrator's policy; (b) a confounded Gopher game where the tunnel and score are masked; (c) a saliency map of `Causal-DQN`'s policy.

On the other hand, in the confounded Gopher game, the gophers' locations are now masked, and the agent no longer follows the same "proactive" strategy. Instead, our `Causal-DQN` picks up an alternative "reactive" strategy, which will reset the farmer's position around the center and only move when a gopher is digging out of the ground. We evaluate the learner's performance and provide them in Table 1 and Fig. 8. Interestingly, simulation results show that the "reactive" strategy is more effective than a "proactive" one, and `Causal-DQN` is able to outperform the demonstrator's policy.

**Confounded ChopperCommand.** In the ChopperCommand game, the agent controls a helicopter tasked with defending a convoy of trucks from waves of enemy aircraft and helicopters. The agent must navigate across the desert landscape, shooting down enemies while avoiding incoming fire. Successfully protecting the convoy and eliminating threats increases the player's score, while

Table 1: Average evaluation returns of agents on the 12 confounded Atari games trained with 1M environment steps and aggregated normalized returns concerning the demonstrator's performance. Bold numbers indicate the best-performing methods. All results are averaged over 5 seeds except that column Random is from [2]. `Causal-DQN` significantly outperforms other DQN baselines.

| Game | Demonstrator | Random | Interv. DQN | Conf. DQN | Conf. LSTM-DQN | `Causal-DQN` (**ours**) |
|---|---|---|---|---|---|---|
| Amidar | 232.4 | 5.8 | 44.0 | 37.8 | 59.0 | **282.6** |
| Asterix | 3080.6 | 210.0 | 650.0 | 429.0 | 479.0 | **2587.0** |
| Boxing | 89.0 | 0.1 | -0.62 | -9.8 | -6.9 | **71.5** |
| Breakout | 219.2 | 1.7 | 2.2 | 1.2 | 4.9 | **131.2** |
| ChopperCommand | 1280.0 | 811.0 | 1192.0 | 1076.0 | 1116.0 | **1658.0** |
| Gopher | 5480.6 | 257.6 | 288.8 | 752.0 | 485.6 | **7327.2** |
| KungFuMaster | 35400.0 | 258.5 | 12416.0 | 13674.0 | 6526.0 | **44196.0** |
| MsPacman | 2316.8 | 307.3 | 1191.6 | 881.8 | 787.4 | **1747.6** |
| Pong | 20.8 | -20.7 | -20.8 | -20.8 | -20.4 | **21.0** |
| Qbert | 4420.6 | 163.9 | 322.5 | 208.5 | 253.5 | **4458.5** |
| RoadRunner | 16560.6 | 11.5 | 1154.0 | 1168.0 | 484.0 | **27414.0** |
| Seaquest | 1412.4 | 68.4 | 237.2 | 281.6 | 164.8 | **980.0** |
| Normalized Mean (↑) | 1.00 | 0.00 | 0.13 | 0.10 | 0.09 | **1.04** |
| Normalized Median (↑) | 1.00 | 0.03 | 0.13 | 0.14 | 0.10 | **1.01** |
| Normalized IQM (↑) | 1.00 | 0.03 | 0.13 | 0.13 | 0.11 | **1.02** |

allowing enemy fire to destroy the trucks results in lost points or lives. At the bottom of the screen, a mini-map/radar is showing incoming trucks and enemies. Fig. 5a shows the saliency map for the demonstrator's policy. It learns to utilize the radar information to "look ahead."

We next consider a confounded Chopper-Command game where the chopper loses its radar and other sensor devices. As a result, the mini-map area, current score, and remaining lives are masked (as shown in Fig. 6b). By applying `Causal-DQN`, the agent picks up a more "spontaneous" playing style, focusing on staying alive and eliminating opponents upfront. Fig. 6c describes a saliency map of this "spontaneous" policy where only nearby opponents are highlighted. Simulation results in Table 1 and Fig. 8 reveal that `Causal-DQN` outperforms other baselines significantly and even surpasses the demonstrator's policy despite having a simpler neural network architecture.

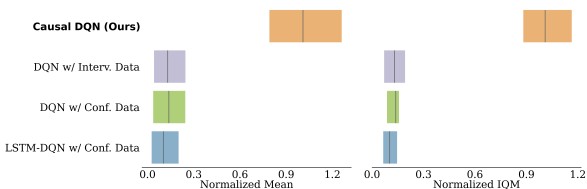

(a)          (b)          (c)

Figure 6: (a) A saliency map of the demonstrator's policy; (b) a confounded ChopperCommand game with the minimap and score/lives masked; (c) a saliency map of `Causal-DQN`'s policy.

**Overall Performance.** Table 1 provides the best mean returns for all 12 confounded Atari games across trials, along with mean return normalized by demonstrator's performance and normalized interquantile mean (IQM). We see `Causal-DQN` consistently outperforming other non-causal baselines by a big margin. In 7/12 games, our proposed method even surpasses the demonstrator with full observations and a way more complex architecture [2]. This could be due to the reason that the demonstrator, though powerful in representation learning, may suffer from observational overfitting [90] and rely on spurious visual features to make decisions. Both prior work and our work have empirically verified this by using saliency maps [25]. Masking out those spurious features indeed poses a non-trivial challenge to the DQN. From Table 1, we see that the vanilla DQN with interventional data under masked observations (Interv. DQN) hardly achieves any meaningful scores in 1 million environment steps. Even with the help of LSTM cells or confounded data from a performing demonstrator, the DQN still cannot learn. Only with the causal bound, the same architecture (Nature DQN) can recover or even surpass the demonstrator's performance despite using masked observations and a shallow, small CNN as the feature extractor. In Fig. 7, we also

Figure 7: Normalized mean and normalized IQM scores. `Causal-DQN` achieves a normalized mean return of 1.04 and a normalized IQM of 1.02.

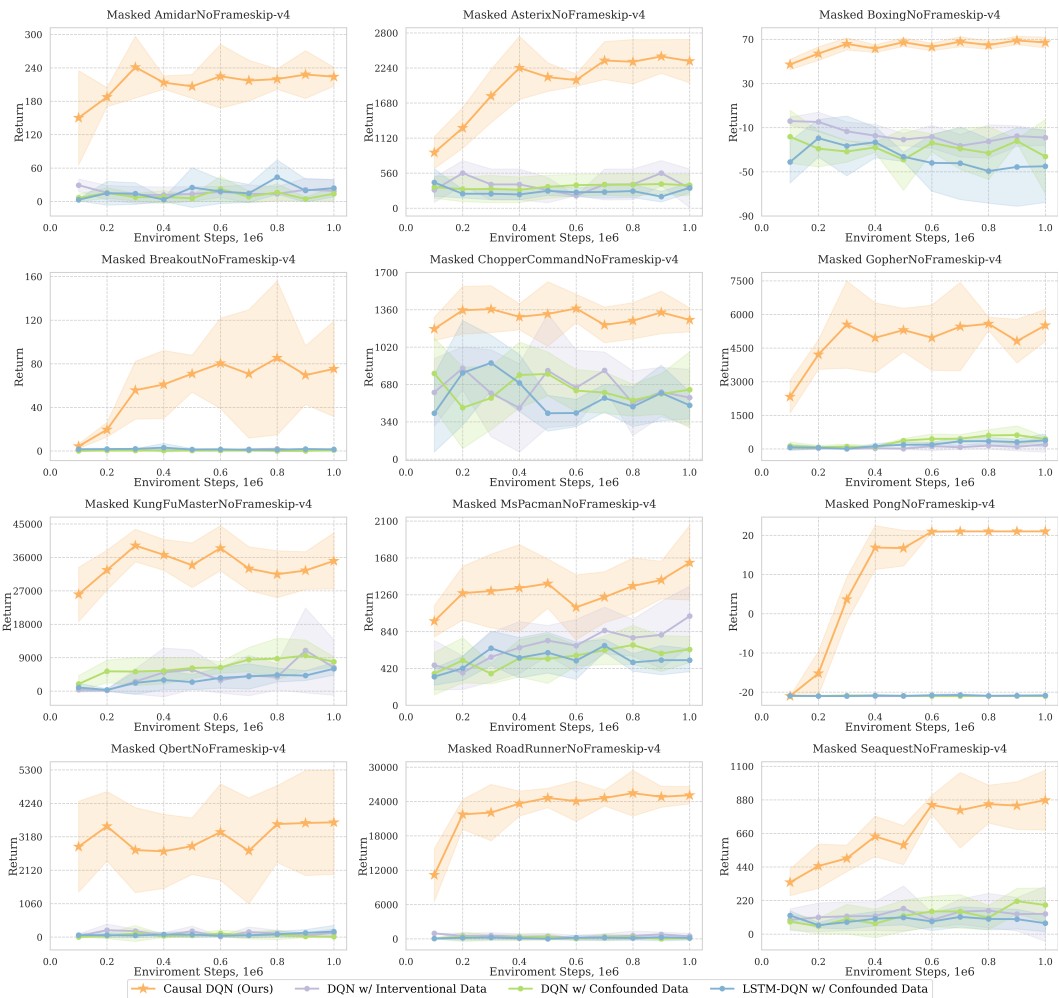

Figure 8: Average evaluation performance for 12 confounded Atari games. During training over the 1M environment steps, we evaluate the agent every 100K steps for 10 episodes each time. The curve is further averaged over 5 seeds with one standard deviation across trials as the shaded area.

provide stratified bootstrap confidence intervals for the normalized mean and normalized IQM scores as recommended by Agarwal et al. [1]. We can see clearly that our proposed `Causal-DQN` can recover the expert demonstrator's performance even under impaired sensors. And to get a sense of the sample efficiency of `Causal-DQN`, in Fig. 8, we report the evaluation performance during training of all 12 confounded Atari games. `Causal-DQN` converges uniformly in less than 1 million steps in all games. See App. E for more results.

**Remark on the failure of DQN baselines.** We have conducted thorough hyper-parameter tuning and used the recommended hyper-parameter settings from the original DQN work [61]. Due to limited time and computational resources, we are only able to finish 1M steps of training for all the environments and seeds. However, we do notice that the interventional DQN baseline can gradually learn the right policy with longer training time. The learning curve usually starts to grow after 3M steps. This corroborates the validity of our confounded environment design, that the learning task is now harder to solve, but not totally impossible. With the help of a causally aware learner (`Causal-DQN`), one would be able to learn more sample-efficiently (3x fewer steps in our experiments) than the pure interventional online regime or the biased causally unaware offline learners.

# 5  Conclusions

This paper investigates deep reinforcement learning from off-policy data collected by a different behavior policy through a causal lens. Particularly, we focus on a generalized setting where confounding biases cannot be ruled out *a priori*, which poses significant challenges to standard off-policy evaluation algorithms. We first extend the celebrated Bellman equation to the causal Bellman equation that lower bounds the agent's expected return from confounded observations. Building on this extension, we then propose a novel `Causal-DQN` algorithm that could obtain an effective policy from off-policy data even when unobserved confounders generally exist. Finally, we evaluate our proposed algorithm in twelve confounded Atari games, showing that the causal approach consistently dominates the standard DQN algorithm with different feature extractors or data sources.

Yet the implications of this work extend far beyond discrete control benchmarks. Unobserved confounding is not an anomaly but a pervasive property of real-world RL. It lurks beneath virtually all forms of observational or off-policy data, from robotic demonstrations to human feedback, silently distorting the mapping between actions, rewards, and outcomes. In an era when the field is increasingly guided by the scaling law, the belief that enlarging models and datasets will automatically yield better intelligence, our findings reveal a critical blind spot: scaling on confounded data does not scale performance. In fact, it may amplify biases, producing policies that are efficient yet misaligned.

Consider large-scale robotic pretraining from internet videos: behavioral policies (human or robot demonstrators) differ sharply from the learner's policy space, introducing systematic unobserved confounding between observations and intended actions. Similarly, in reinforcement learning from human feedback (RLHF) for aligning large language models (LLMs), preference datasets are deeply entangled with hidden factors like emotions, social norms, cultural context, temporal inconsistency, and individual baselines that no simple text prompt can fully encode. Without a causal understanding of these factors, LLM alignment merely approximates correlations within preference and prompts, not their causal origins. The same problem is exacerbated for healthcare, finance, and any domain where observational data conceals the determining forces driving decisions.

Toward the future, we envision causal reinforcement learning as a foundation for building confounding-robust, safely-aligned, and generalizable agents. Extending the causal Bellman framework to policy-gradient methods, continuous control, RLHF, and multi-agent systems will be crucial steps toward this goal. Ultimately, bridging causal inference and deep RL offers more than robustness. It paves the way for agents that reason about interventions and consequences, rather than merely fitting experience. In sum, this work marks an early yet essential stride toward causally grounded agents, a future where RL agents learn not only what to do, but why it works.

## Acknowledgments

This research is supported in part by the NSF, ONR, AFOSR, DoE, Amazon, JP Morgan, and The Alfred P. Sloan Foundation.

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

# Appendices

**Contents**

## A   Related Work

**Off-Policy Learning.** Off-policy learning has a long history in RL dating back to the classic algorithms of Q-learning [105, 106], importance sampling [38, 94], and temporal difference [65, 76]. Recently, people also propose to utilize offline datasets to warm start the training [16, 50, 66], augmenting online training replay buffer [8, 91] or incorporating imitation loss with offline data [45, 116]. However, these work rely on a critical assumption that there is no unobserved confounders in the environment. While this assumption is generally true when the off-policy data is collected by an interventional agent, data generated by potentially unknown sources can easily break this assumption [53]. We will introduce more on the confounding robust off-policy learning next.

**Causal Reinforcement Learning for Off-Policy Learning** When the no unobserved confounding assumption does not hold, one would need to either identify the reward and transition distributions before evaluating policy values or bound the possible policy values. There is a rich line of literature in identifying policy values directly from confounded data [13, 27, 60, 87]. But they usually invoke other critical learning assumptions such as the existence of bridge functions in the line of proximal causal inference literature [96]. On the other hand, without further assumptions, one can utilize the bounding method to account for the whole range of possible policy values. Seminal work of Manski [59] developed the first bounds on causal effects in non-identifiable settings using observational data in the single-stage treatment model with contextual information (i.e., a contextual bandit model). These bounds were then expanded to the instrumental variable setting [7, 34], to partially identify counterfactual probabilities of causation [98], to construct reward shaping functions automatically [54]. This work is inspired by a recent work of Zhang & Bareinboim in partially identifying the policy values via bounding [113].

**Robust Reinforcement Learning** Unlike reinforcement learning in standard MDPs, robust reinforcement learning assumes that the parametrization of the transition probability function is contained in a set of model parameters which is called the uncertainty set [36, 56, 68, 73, 107–109]. The goal of the agent is to learn a robust policy that performs the best under the worst possible case in the uncertainty set. Similar problems have been studied under the rubrics of safe policy learning [23, 97] or pessimistic reinforcement learning [58]. Robust RL algorithms with provable guarantees have been proposed in tabular settings or under the assumptions of linear functions [5, 55, 80, 95, 102]. Combined with the computational framework of deep learning, robust RL algorithms have been extended to complex, high-dimensional domains [74, 110]. More recently, [71] proposed Robust Fitted Q-Iteration (RFQI) to learn the best possible robust policy from offline data with theoretical guarantees on the performance of the learned policy. Our work differs from robust RL methods since it does not require a pre-specified uncertainty set of model parameters. Instead, we construct the ignorance region over the underlying system dynamics from the confounded observational data

using partial causal identification. Based on the learned uncertainty set, we then derived closed-form bounds over the optimal interventional Q-value functions.

**Model-Free Deep Reinforcement Learning** Model-free deep reinforcement learning has seen substantial advancements through the evolution of value-based methods, particularly those building on the foundational Deep Q-Network (DQN)[61]. While DQN introduced the use of deep neural networks to approximate Q-values from high-dimensional sensory inputs, it suffered from issues such as overestimation bias, instability, and poor data efficiency. These challenges prompted a series of algorithmic improvements. Double DQN [101] addresses overestimation by separating action selection from action evaluation, while Dueling DQN [103] improves representational efficiency by decoupling value and advantage estimation. Prioritized Experience Replay [82] and Hindsight Experience Replay [3] enhances data efficiency by sampling more informative transitions. Rainbow DQN [31] effectively combines these ingredients, along with multi-step learning, distributional Q-learning, and noisy networks, resulting in one of the strongest baselines in Atari environments.

To scale value-based methods to more complex tasks and hardware infrastructures, subsequent works introduced distributed and more flexible architectures. IMPALA (Importance Weighted Actor-Learner Architectures) [21] tackles the inefficiency of distributed policy learning by decoupling acting and learning processes and correcting the resulting off-policy updates with the V-trace algorithm, enabling efficient parallel training at scale. R2D2 (Recurrent Experience Replay in Distributed Reinforcement Learning) [46] enables recurrent architectures to be trained off-policy in distributed settings with experience replay, supporting partial observability and long time horizons. Agent57 [4] further builds on R2D2, combining exploration bonuses from Random Network Distillation [19], meta-learning of exploration strategies, and population-based training, becoming the first agent to surpass human performance on all 57 Atari games. More recently, the Bigger, Better, Faster (BBF) framework [86] pushes the scalability frontier by optimizing infrastructure, neural network design, and training pipelines, enabling the training of large-scale agents with dramatically improved performance and sample efficiency. These innovations collectively mark a significant advancement toward more scalable and generalizable value-based deep reinforcement learning agents.

Note that our proposed Causal Bellman Optimality Equation does not require specific reinforcement learning algorithm implementations. In this work, we choose DQN as the base algorithm only for its simplicity given our goal of showcasing a practical implementation of the proposed result in a straightforward way. It is possible to extending the causal Q-learning update in Eq. (8) to be used with more advanced deep RL algorithms (e.g., Rainbow [31], IMPALA [21], and BBF [86]) so that one could enable more powerful agents in confounding settings, which we will leave for future work.

# B  Proof Details

This section entails the proof for Prop. 3.1. We also prove that our Causal Bellman Optimal Equation (lower bound) has a unique fixed point that is a valid lower bound.

**Proposition B.1** (Causal Bellman Optimal Equation (Prop. 3.1)). *For a CMDP environment $\mathcal{M}$ with reward signals $Y_t \in [a, b] \subseteq \mathbb{R}$, its optimal state-action value function $Q_*(s, x) \geq \underline{Q}_*(s, x)$ for any state-action pair $(s, x) \in \mathcal{S} \times \mathcal{X}$, where the lower bound $\underline{Q}_*(s, x)$ is given by as follows,*

$$\underline{Q}_*(s, x) = P(x \mid s)\left( \widetilde{\mathcal{R}}(s, x) + \gamma \sum_{s', x'} \widetilde{\mathcal{T}}(s, x, s') \max_{x'} \underline{Q}_*(s', x') \right) \tag{12}$$

$$+ P(\neg x \mid s)\left( a + \gamma \min_{s'} \max_{x'} \underline{Q}_*(s', x') \right) \tag{13}$$

*where $P(x \mid s) = P(X_t = x \mid S_t = s)$ and $P(\neg x \mid s) = 1 - P(x \mid s)$; $\widetilde{\mathcal{T}}$ and $\widetilde{\mathcal{R}}$ are nominal transition distribution and reward function computed from the observational distribution, i.e.,*

$$\widetilde{\mathcal{T}}(s, x, s') = P(S_{t+1} = s' \mid S_t = s, X_t = x), \qquad \widetilde{\mathcal{R}}(s, x) = \mathbb{E}[Y_t \mid S_t = s, X_t = x] \tag{14}$$

*Proof.* Starting from the Bellman Optimal Equation for Q-values, the optimal state action value function is given by,

$$Q^*(s, x) = R(s, x) + \sum_{s'} T(s, x, s') \max_{x'} Q^*(s', x') \tag{15}$$

Note that the actions in the reward and transition functions are done by an interventional agent, which is actually $\text{do}(x)$ in the context of a CMDP. Due to the confounding nature of those two distributions, we can use the natural bounds to bound the interventional reward ($\mathcal{R}$) and transition distribution ($\mathcal{T}$) with observational data ($\widetilde{\mathcal{R}},\widetilde{\mathcal{T}}$) [59].

$$\mathcal{R}(s,x) \geq \widetilde{R}(s,x)P(x|s) + aP(\neg x|s) \tag{16}$$

$$\sum_{s'} T(s,x,s')\max_{x'} Q^*(s',x') \geq \sum_{s'} \widetilde{T}(s,x,s')P(x|s)\max_{x'} Q^*(s',x')$$
$$+ P(\neg x|s)\min_{s'}\max_{x'} Q^*(s',x') \tag{17}$$

Then we have,

$$Q^*(s,x) \geq \widetilde{R}(s,x)P(x|s) + aP(\neg x|s) +$$
$$\sum_{s'} \widetilde{T}(s,x,s')P(x|s)\max_{x'} Q^*(s',x') + P(\neg x|s)\min_{s'}\max_{x'} Q^*(s',x') \tag{18}$$

where $\widetilde{\mathcal{R}}_h(s,x) = \mathbb{E}[Y_h|S_h = s, X_h = x]$, $\widetilde{\mathcal{T}}_h$ is shorthand for $\widetilde{\mathcal{T}}_h(s,x,s') = P(S_{h+1} = s'|S_h = s, X_h = x)$ and $P(x|s) = P_h(X_h = x|S_h = s)$ are estimated from the offline dataset. And $a$ is a known lower bound on the reward signal, $Y_h \leq b$. In this step, we lower bound the next state transition by assuming the worst case that for the action not taken with probability $P_h(\neg x|s)$, the agent transits with probability 1 to the worst possible next state, $\min_{s''} V_{h+1}^*(s'')$.

Then after rearranging terms, we have,

$$Q^*(s,x) \geq P(x \mid s)\left(\widetilde{\mathcal{R}}(s,x) + \gamma \sum_{s',x'} \widetilde{\mathcal{T}}(s,x,s')\max_{x'} Q^*(s',x')\right)$$
$$+ P(\neg x \mid s)\left(a + \gamma \min_{s'}\max_{x'} Q^*(s',x')\right) \tag{19}$$

Optimizing the Q-value function w.r.t. this inequality gives us a lower bound on the optimal state value. Replace the symbol $Q^*$ with $\underline{Q}_*$ and we have,

$$\underline{Q}_*(s,x) \geq P(x \mid s)\left(\widetilde{\mathcal{R}}(s,x) + \gamma \sum_{s',x'} \widetilde{\mathcal{T}}(s,x,s')\max_{x'} \underline{Q}_*(s',x')\right)$$
$$+ P(\neg x \mid s)\left(a + \gamma \min_{s'}\max_{x'} \underline{Q}_*(s',x')\right) \tag{20}$$

$$\square$$

Next, we will show in Prop. B.2 that this will converge to a unique fixed point, which is a valid lower bound of the optimal state-action value function.

**Proposition B.2** (Convergence of Causal Bellman Optimal Equation in Stationary CMDPs). *The Causal Bellman Optimality Equation converges to a unique fixed point, which is also a lower bound on the optimal interventional state values under the assumption that in the observational data $P(s,x) > 0, \forall s, x$ in the given CMDP.*

*Proof.* We will first show that the following Causal Bellman Optimality operator (will denote as "the operator" or $T$ below for simplicity) is a contraction mapping with respect to a max norm. Then by Banach's fixed-point theorem [9], this operator has a unique fixed point, and updating any initial point iteratively will converge to it. Then we show that this unique fixed point is indeed a lower bound of the optimal interventional Q-value.

Let the operator $T$ be,

$$T\underline{Q}^*(s,x) = P(x \mid s)\left(\widetilde{\mathcal{R}}(s,x) + \gamma \sum_{s',x'} \widetilde{\mathcal{T}}(s,x,s')\max_{x'} \underline{Q}_*(s',x')\right)$$
$$+ P(\neg x \mid s)\left(a + \gamma \min_{s'}\max_{x'} \underline{Q}_*(s',x')\right). \tag{21}$$

For arbitrary Q-value bound, $\underline{Q}_*^1, \underline{Q}_*^2$, let their initial difference under max-norm be $c = \max_{s,x} \left| \underline{Q}_*^1(s,x) - \underline{Q}_*^2(s,x) \right| \geq 0$. We can bound their difference after one step update by,

$$\max_{s,x} \left| T\underline{Q}_*^1(s,x) - T\underline{Q}_*^2(s,x) \right| \leq \gamma \max_{s,x} \left| P(x|s) \sum_{s'} \widetilde{T}(s,x,s') \max_{x'} \left| \underline{Q}_*^1(s',x') - \underline{Q}_*^2(s',x') \right| \right.$$
$$\left. + P(\neg x|s) \max_{s',x'} \left| \underline{Q}_*^1(s',x') - \underline{Q}_*^2(s',x') \right| \right|. \tag{22}$$

Thus, under the operator $T$, we have non-expansion Q-value differences,

$$\max_{s,x} \left| T\underline{Q}_*^1(s,x) - T\underline{Q}_*^2(s,x) \right| \leq \gamma \max_{s,x} \left( P(x|s) \sum_{s'} \widetilde{T}(s,x,s') \max_{x'} \left| \underline{Q}_*^1(s',x') - \underline{Q}_*^2(s',x') \right| \right.$$
$$\left. + P(\neg x|s) \max_{s',x'} \left| \underline{Q}_*^1(s',x') - \underline{Q}_*^2(s',x') \right| \right), \tag{23}$$

$$\leq \gamma c \max_{s,x} \left( P(x|s) \sum_{s'} \widetilde{T}(s,x,s') + P(\neg x|s) \right), \tag{24}$$

$$= \gamma c. \tag{25}$$

for all $\underline{Q}_*^1, \underline{Q}_*^2$ satisfying $c \geq \max_{s,x} \left| \underline{Q}_*^1(s,x) - \underline{Q}_*^2(s,x) \right| \geq 0$. Thus, $T$ is a contraction mapping with respect to the max norm. And there exists a unique fixed point $\underline{Q}_*$ when we apply this operator $T$ iteratively to an arbitrary Q-value vector till convergence.

We then show that this fixed point is indeed a lower bound to the optimal interventional Q-value. By the update rule of $T$ (Eq. (21)), $\forall Q(s,x), Q(s,x) \geq TQ(s,x)$. Thus, for the optimal Q-value, we can have $Q^*(s,x) \geq \lim_{k \to \infty} T^k Q^*(s,x) = \underline{Q}_*(s,x)$ where $T^k$ denotes applying $T$ iteratively for $k$ times. This concludes the proof. □

## C  Confounded Atari Games Design

In this section, we present the detailed design of each confounded Atari games. The core design idea is that we would like to occlude the part of the screens that contains information useful for making decisions but is not a significant factor from human players' perspectives. For example, the remaining lives and current scores can be an indicator of the difficulty level or even a unique game level identifier. However, such information is not usually exploited intensively in human game plays. Thus, we decide to mask out such regions.

Below is a detailed list of confounders design for each game.

- **Amidar**: The original game screen shows score and remaining life which shouldn't be the major factor affecting the policies.
- **Asterix**: The original game screen shows score and remaining life which shouldn't be the major factor affecting the policies.
- **Boxing**: The original game screen shows score and remaining time which shouldn't be the major factor affecting the policies. The right half of the arena can also be excluded since there is still a wining strategy by staying on the left hand side (as long as the demonstrator has such demonstrations on the left hand side).
- **Breakout**: The original game screen shows score and remaining life which shouldn't be the major factor affecting the policies.
- **ChopperCommand**: The original game screen shows score and remaining life which shouldn't be the major factor affecting the policies. The mini map can be helpful, but without it, there should also be a good policy.
- **Gopher**: The original game screen shows score and remaining life which shouldn't be the major factor affecting the policies. The gopher location, while nice to have, can be removed to force the learning to focus more on the hole not the gopher.
- **KungFuMaster**: The original game screen shows score and remaining life which shouldn't be the major factor affecting the policies.

- **MsPacman**: The original game screen shows score and remaining life which shouldn't be the major factor affecting the policies.
- **Pong**: The original game screen shows score and remaining life which shouldn't be the major factor affecting the policies. Moreover, one can simply win the game by shoot back every incoming ball. Thus, we can further block out the opponent's paddle location.
- **Qbert**: The color itself is already a good confounder. The demonstrator model [2] is observing three channel RGB images with LSTM cells. The student model only has grayscale image stacks. See more details in App. E.
- **RoadRunner**: The original game screen shows score and remaining life which shouldn't be the major factor affecting the policies. Also the sky/desert part are mostly
- **Seaquest**:The original game screen shows score and remaining life which shouldn't be the major factor affecting the policies.

We show each masked atari games in Fig. 9.

## D   Implementation Details and Experiment Setups

The input to all the networks consists of a stack of four grayscale frames with a frame skipping of four (so the agent observes a stack of four out of sixteen consecutive actual frames), each down sampled to 64×64 resolution, allowing the agent to infer temporal dynamics such as ball velocity. To reduce the inherent flickering from the Atari game, we also apply max pooling over each two consecutive actual frames. We also downsize the input to 64x64 to align with the input size requirement of the demonstrator, diamond [2]. The only differences in terms of observations are that (1) the demonstrator takes a single colored frame (also max-pooled) as input per each time step, and (2) the demonstrator has access to the original full screen observation while the learners only has access to a masked partial screen. For the reward, we clip it between $[-1, +1]$ to stabilize training.

For the other demonstrator, we use the sebulba model [30] implemented by the CleanRL package [33]. Its backbone is from Impala [21] architecture and the training algorithm is PPO [85]. Its input image is in 81x81 but still are stacked grayscale images, the same as what the DQN agents use. Actions are selected as the argmax action with the biggest logits after applying the Gumbel softmax trick [37]. The training batch size is also increased to 2048 and is trained with cosine annealing learning rate scheduler. For all runs, the learning rate is set to 5e-4. For the consine annealing learning rate scheduler, the minimum learning rate is 1e-6.

For all CNN based DQN networks in this work, we adopt the nature DQN architecture introduced by Mnih et al. [61]. The network comprises three convolutional layers followed by two fully connected layers, outputting Q-values for each discrete action. While for LSTM based ones, we only replace the second to last linear layer in nature DQN with an LSTM cell. For both the linear layers and lstm cells, we use a hidden dimension of 512. To mitigate overestimation bias in Q-learning, we further incorporate the Double DQN modification [101], which decouples action selection and evaluation in the target update by using the online network to select actions and the target network to evaluate their value. This leads to more stable and accurate value estimates. We also use epsilon greedy for exploration as the standard DQN algorithm. See Algo. 2 for the full pseudo-code.

To train our model, we use an H100 GPU. On average, for each game and each seed, it takes around 2 hrs and a RAM space of less than 2 GB for using the diamond demonstrator [2]. While it takes up to 8 hrs for using the sebulba demonstrator [30] from CleanRL [33].

## E   More Experiment Results

Here we present the experiment results for agents trained with another even stronger demonstrator, sebulba [30]. From Table 2 and Fig. 10, we can draw the same conclusion that our `Causal-DQN` is robust to confounded demonstrator demonstrations and is able to extract useful policies out of such demonstrations. And we can see that in most of the games, the agent trained by sebulba demonstrator significantly outperforms the agent trained by diamond demonstrator [2]. This can be an empirical evidence that our proposed `Causal-DQN` is able to scale with the demonstrator's performance. But his also reveals two limitations of our current approach that when the demonstrator generated data has zero support in the unmasked area, it is not possible to learn any meaningful behaviors from such

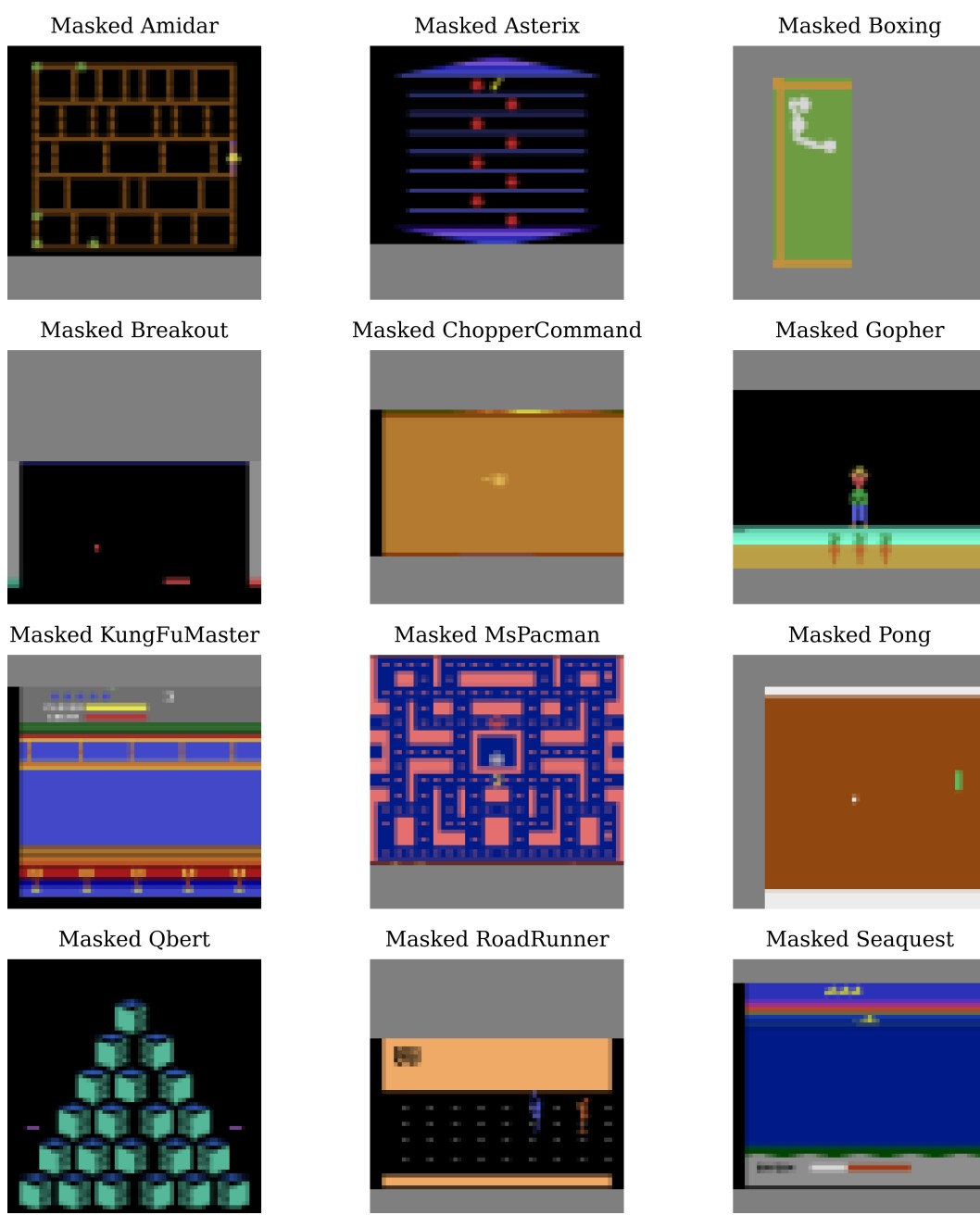

Figure 9: All 12 confounded Atari games. Masked areas are shown in grey.

demonstrations. In our current setup of the confounded Boxing game, we mask out the right half but the sebulba demonstrator has a policy of fighting in the right half. Thus, none of the algorithms we tested can learn. To further verify this intuition, we mask out the left hand side of the arena in the Boxing game (App. E) and rerun all baselines and `Causal-DQN`. As expected, as shown in App. E, our model is now able to converge to the optimal 100 score matching the demonstrator's performance while other baeslines still struggle to learn from the confounded demonstrations. Also, when the demonstrator's policy is distributional and multi-modal, i.e., there could be multiple best actions, the deterministic policy like DQN cannot capture such knowledge very well. In Asterix, due to the non-standard way of using Gumbel softmax implemented by CleanRL [33], there could be different optimal actions for the same state, posing a challenge to the deterministic learners.

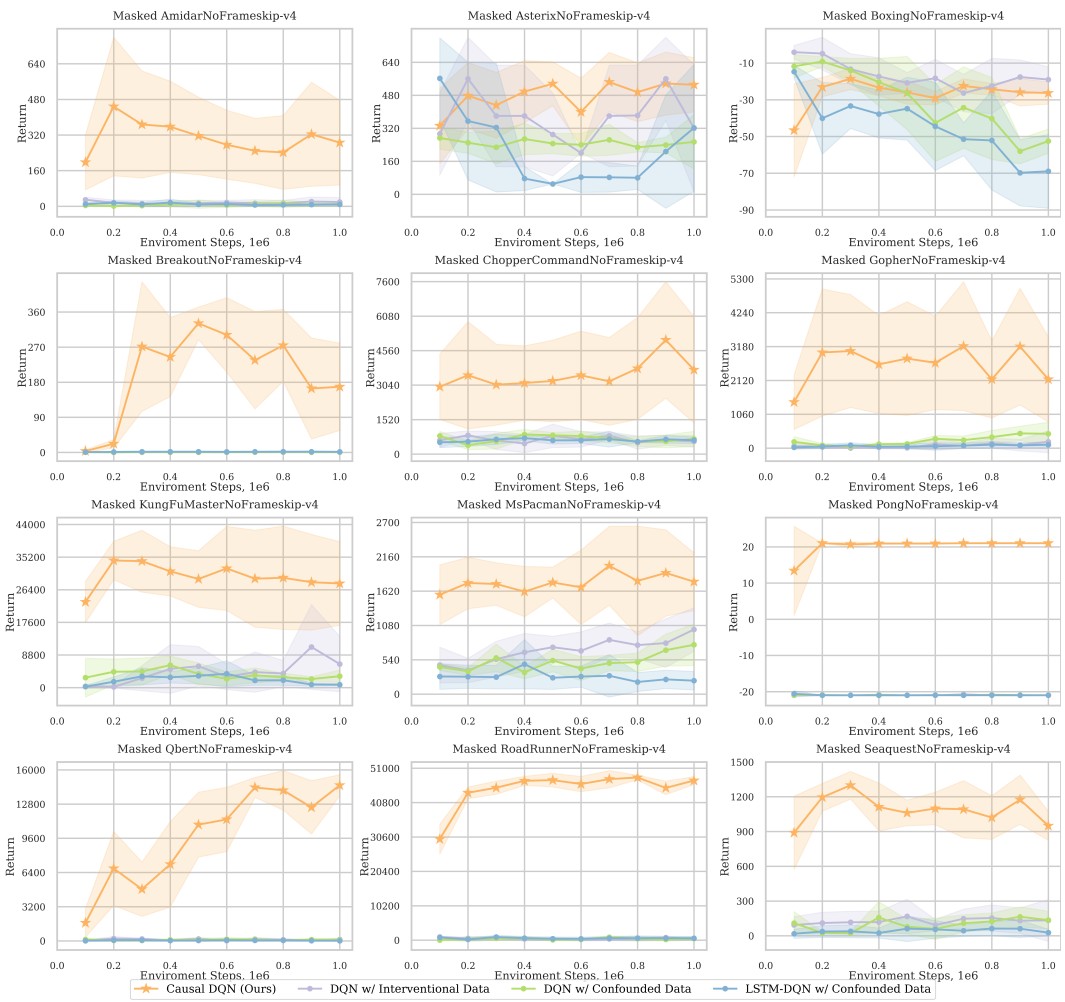

Figure 10: Average evaluation performance for 12 confounded Atari games with sebulba [30] as the demonstrator. During training over the 1M environment steps, we evaluate the agent every 100K steps for 10 episodes each time. The curve is further averaged over 5 seeds with one standard deviation across trials as the shaded area.

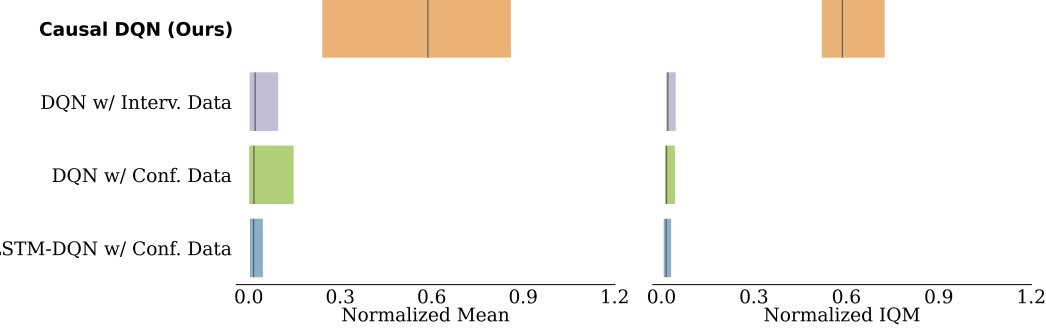

Figure 11: Normalized mean and normalized IQM scores. `Causal-DQN` achieves a normalized mean return of 0.55 and a normalized IQM of 0.59 with sebulba demonstrator.

**Algorithm 2** Causal Deep Q-Learning (`Causal-DQN`)

---

1: Initialize replay memory $\mathcal{D}$
2: Initialize action-value function $\underline{Q_*}^\theta$ and a target network $\underline{Q_*}^{\theta^-}$, $\theta^- \leftarrow \theta$
3: **for** episodes $= 1, \ldots, M$ **do**
4:     Sample initial state $s_1$ and obtain preprocessed $\phi_1 = \phi(s_1)$
5:     **for** $t = 1, \ldots, T$ **do**
6:         With probability $\epsilon$ select a random action $x_t$
7:         Otherwise sample an action from demonstrator, $x_t \leftarrow f_X(s_t, u_t)$
8:         Execute action $\text{do}(x_t)$ in environment and observe reward $y_t$ and state $s_{t+1}$
9:         Store transition $(s_t, x_t, y_t, s_{t+1})$ in $\mathcal{D}$
10:       Sample a minibatch of transitions $\{(s_i, x_i, y_i, s_{i+1})\}_{i=1}^B$ from $\mathcal{D}$
11:       Set value target $w_i(x)$ for every action $x \in \mathcal{X}$ w.r.t sample $(s_i, x_i, y_i, s_{i+1})$,

$$w_i(x) = \begin{cases} y_i + \gamma \max_{x'} \underline{Q_*}(s_{i+1}, x'; \theta^-) & \text{if } x = x_i \\ a + \gamma \min_{s'} \max_{x'} \underline{Q_*}(s', x'; \theta^-) & \text{if } x \neq x_i \end{cases} \tag{26}$$

12:       Perform a gradient descent step on $\sum_x \left( w_i(x) - \underline{Q_*}(s_i, x; \theta) \right)^2$ according to Eq. (10)
13:       Every $T_{\text{target}}$ steps, update $\theta^- \leftarrow \theta$
14:     **end for**
15: **end for**

---

Table 2: Average evaluation returns of agents on the 12 confounded Atari games trained with 1M environment steps and aggregated normalized returns concerning the sebulba demonstrator's performance [30]. Bold numbers indicate the best-performing methods. All results are averaged over 5 seeds except that column Random is from [2]. `Causal-DQN` significantly outperforms others.

| Game | Demonstrator (sebulba) | Random | Interv. DQN | Conf. DQN | Conf. LSTM-DQN | Causal-DQN (diamond) | Causal-DQN (sebulba) |
|---|---|---|---|---|---|---|---|
| Amidar | 2148.2 | 5.8 | 44.0 | 22.6 | 24.6 | 282.6 | **462.8** |
| Asterix | 250182.0 | 210.0 | 650.0 | 369.0 | 662.0 | **2587.0** | 586.0 |
| Boxing | 100.0 | 0.1 | -0.62 | -7.6 | -13.4 | **71.5** | -16.86 |
| Breakout | 771.0 | 1.7 | 2.2 | 0.9 | 2.9 | 131.2 | **408.2** |
| ChopperCommand | 21682.0 | 811.0 | 1192.0 | 1096.0 | 918.0 | 1658.0 | **5410.0** |
| Gopher | 3719.2 | 257.6 | 288.8 | 646.4 | 132.0 | **7327.2** | 4008.0 |
| KungFuMaster | 46046.0 | 258.5 | 12416.0 | 8468.0 | 4844.0 | **44196.0** | 39222.0 |
| MsPacman | 4538.4 | 307.3 | 1191.6 | 963.6 | 561.4 | 1747.6 | **2346.8** |
| Pong | 21.0 | -20.7 | -20.8 | -20.8 | -20.6 | **21.0** | 21.0 |
| Qbert | 23484.0 | 163.9 | 322.5 | 283.5 | 136.5 | 4458.5 | **15136.0** |
| RoadRunner | 56056.0 | 11.5 | 1154.0 | 1182.0 | 1108.0 | 27414.0 | **49482.0** |
| Seaquest | 1797.2 | 68.4 | 237.2 | 247.6 | 101.6 | 980.0 | **1350.0** |
| Normalized Mean (↑) | 1.00 | 0.00 | 0.00 | 0.00 | 0.00 | 0.53 | **0.55** |
| Normalized Median (↑) | 1.00 | 0.01 | 0.03 | 0.04 | 0.02 | 0.40 | **0.59** |
| Normalized IQM (↑) | 1.00 | 0.0 | 0.02 | 0.02 | 0.02 | 0.47 | **0.59** |

# F   Broader Impact

This paper presents work whose goal is to advance the field of Reinforcement Learning. There are many potential societal consequences of our work, none of which we feel must be specifically highlighted here. One major reason is that our proposed algorithm aims at extracting knowledge from another demonstrator model solving Atari games of which we don't find any profound social impacts worth mentioning here.

# G   Limitations

Our current derivation applies to single step Q-value functions. For other objectives for critics like multi-step returns, eligibility traces and advantages, we need to further extend the Causal Bellman Equation to accommodate those. As we also show in App. E, the proposed `Causal-DQN` cannot learn useful policies when the demonstrator policy has no support in the unmasked area. For example, in the boxing game, we mask out the right half of the screen while the demonstrator agent's winning

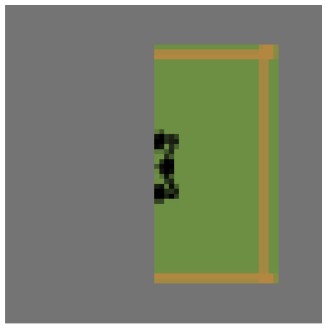 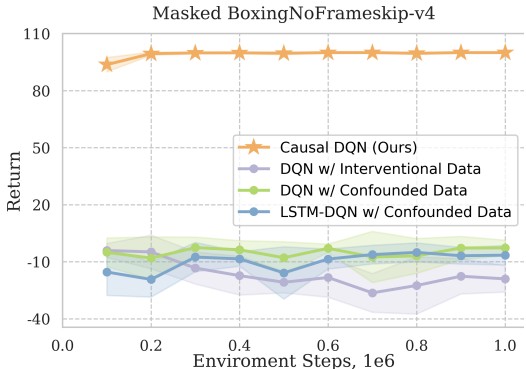

Figure 12: Left: The confounded Boxing game with left side arena masked. Right: Evaluation returns of `Causal-DQN` and other baselines. The curve is an average over five random seeds with one standard deviation as the shaded area. The orange curve is our `Causal-DQN` and other colors are baselines. Same legend following previous figures.

policy is to fight in the right half. In which case, none of the agent under masked observations is able to learn. Furthermore, in games like Asterix where a distributional demonstrator has a more stochastic behavior, our `Causal-DQN` also cannot outperform others. We hypothesis this to be an inherent representational limit of deterministic DQN policies.

