# OpenReview forum: "Confounding Robust Deep Reinforcement Learning: A Causal Approach"
_NeurIPS.cc/2025/Conference — NeurIPS 2025 poster_

### Official Review · Reviewer_E4zm · 2025-06-18

**Clarity:** 3
**Significance:** 3
**Originality:** 3
**Rating:** 4
**Confidence:** 2

**Summary:**

This paper introduces Causal-DQN, a novel reinforcement learning algorithm designed to be robust against unobserved confounding in offline, high-dimensional environments. This approach specifically tackles the scenarios when the observed action and reward are confounded. The mothod first incorporates  a Causal Bellman optimality equation that provides a lower bound on the value function, and then designs an algorithm that minimizes this lower bound and demonstrates its effectiveness across 12 carefully Atari environments.

**Questions:**

1. Did the author discover any chance that the policy behaves too conservative? Given that a lower bound is learned. In particular,  I think it would be helpful to discuss under what scenarios that pessimism is likely to happen or not.
2. Regarding the assumptions:
   * The authors first assume the action domain to be discrete while the state domain to be continuous. Could the method be extend to continuous action space?
   * The authors assume that there's unobserved confounder $U_t$ that affects the action, state and reward simultaneously. Could the authors provide motivation for dealing with this type of confounding? For example, I think in the Pong setting, the confounder (the opponent's position) seems to be a cause for the action and outcome, but the next state should be generated from a transition function that follows physical laws regardless of $U_t$. Why considering the transition function as $S\times X \times U \rightarrow S$ instead of  $S\times X  \rightarrow S$. It is also slightly confusing to use three separate dashed double-arrows to represent the same one $U_t$.

**Ethical Concerns:**

["NO or VERY MINOR ethics concerns only"]

**Final Justification:**

I recognize the novelty and contribution of this work for making a first step towards solving confounding RL problems. I think the author's rebuttal further clarifys their method. I'm not completely sure about I understand all the details but I'm inclining to accept.

**Limitations:**

Yes, the authors has discussed the limitations about this work.

**Quality:**

3

**Strengths And Weaknesses:**

**Quality**

Strengths:

- The theoretical gurantees are sound.
- The algorithm is well-formulated with careful attention to both theory and implementation.
- Experiments are comprehensive with 12 Atari games.



 **Clarity**

Strengths:

- The paper is well written and the logic is easy to follow.
- The Proposition and Algorithm are clearly presented.

Weaknesses:

- The explanation of some derivations (especially in Proposition 3.1) assumes familiarity with causal RL; a brief intuition would help.

 **Significance**

Strengths:

- Tackles an under-explored but important problem in RL: confounding in offline learning.
- Demonstrates improvements in challenging settings where standard DQNs fail.

Weaknesses:

- The scenarios are limited to Atari games; results on more diverse domains would be interesting to be discussed.

**Originality**

Strengths:

- Extending the Bellman equation to the partially identifiable, confounded setting is novel and impactful.

Weaknesses:

- The framework largely builds on prior causal RL work by Zhang & Bareinboim; the originality lies more in algorithmic application.

---

> ### Author Rebuttal · Authors · 2025-07-31
>
> We thank the reviewer for the feedback and appreciate your recognition of the significance of our work. We have addressed each of your concerns in the responses below.
>
> > **W1** “The explanation of some derivations (especially in Proposition 3.1) assumes familiarity with causal RL; a brief intuition would help.”
>
> Intuitively, because of the existence of confounders, we are unsure about whether the worst case can indeed happen in reality or not given solely the observational data. Thus, we compensate the standard Bellman update with the worst case transitions resulting in a pessimistic lower bound to the optimal Q-values. For more details on causal RL, we kindly refer you to [6].
>
> > **W2** “The scenarios are limited to Atari games; results on more diverse domains would be interesting to be discussed.”
>
> We follow a similar evaluation framework to the original DQN, which has been widely applied in many real-world applications. For other domains like those continuous action environments, it is beyond the scope of this work, which we plan to pursue next.
>
> > **W3** “The framework largely builds on prior causal RL work by Zhang & Bareinboim; the originality lies more in algorithmic application.”
>
> Despite algorithmic applications, we would like to point out that on the theoretical side, we propose the first confounding-robust off-policy learning algorithm that bounds the target Q-value using function approximation. The derivation of the learning updates and how we combine them into a sample-based learning method like DQN is non-trivial. Also, Zhang & Bareinboim (2025) only consider the policy evaluation problem, and do not provide policy recommendations.  In this work, we study the more challenging policy learning setting, and Causal-DQN is able to obtain an effective conservative policy from confounded observations.
>
> Empirically, to the best of our knowledge, we are the first to perform confounding robust learning methods for high-dimensional environments (like Atari games) with function approximations, while the previous paper like (Zhang & Bareinboim 2025) only handles small discrete state spaces like very small grid worlds.
>
> > **Q1** “Did the author discover any chance that the policy behaves too conservative? Given that a lower bound is learned. In particular, I think it would be helpful to discuss under what scenarios that pessimism is likely to happen or not. ”
>
> There exist cases where there is a considerable performance gap between our Causal-DQN policy and the demonstrator policy using an LSTM layer. This means that our causal bound could be loose. Still, our conservative policy is able to outperform other standard algorithms that directly use the confounded observations without accounting for the potential confounding bias. It’s possible that we can further improve the bound and be more optimistic with additional domain knowledge or assumptions. But under the general case of CMDP definitions, our Causal Bellman Equation provides a tight lower bound on the optimal Q-values, and thus the policy learned is robust against the worst-case scenarios.
>
> Empirically, we actually observe that sometimes a conservative policy still works well, like in Pong and RoadRunner. There are instances where our method is inferior to the behavioral policy due to conservatism. But from Table 1, we see that on average, the newly proposed Causal-DQN’s normalized mean/median/IQM scores are even slightly better than the original behavioral policy with full observations.
>
> > **Q2-1** “The authors first assume the action domain to be discrete while the state domain to be continuous. Could the method be extended to continuous action space?”
>
> Yes, it is possible. This work is a first step towards handling confounding robust learning in high-dimensional state space. Learning with the continuous action space is an important next step for generalized confounding robust deep RL; thank you for the note.
>
> > **Q2-2** “The authors assume that there's an unobserved confounder $U_t$ that affects the action, state and reward simultaneously. Could the authors provide motivation for dealing with this type of confounding? For example, I think in the Pong setting, the confounder (the opponent's position) seems to be a cause for the action and outcome, but the next state should be generated from a transition function that follows physical laws regardless of $U_t$. Why consider the transition function as $S\times X \times U \mapsto S$ instead of $S\times X \mapsto S$. It is also slightly confusing to use three separate dashed double-arrows to represent the same one .”
>
> We use U to represent things that are not recorded in the dataset. For example, when the behavioral policy is a higher-end robot with more sensory information, we want to use such trajectories to train a commercial robot with fewer sensors. CMDP can then be used to model such scenarios. In the Pong game, the next state also involves the ball locations, and clearly, the opponent’s paddle location will affect the ball locations when it hits/misses the ball. Thus, in this case, there is a causal relationship between the latent opponent’s location ($U_t$) and the next ball placement ($S_t$), and the transition function is in the form of $S\times X \times U \mapsto S$.
>
> Regarding the usage of three separate dashed double-arrows, we follow the standard notations in Causality [[Pearl (2009)]](https://bayes.cs.ucla.edu/BOOK-2K/) to denote confounders in the causal diagram. That is, we draw a dashed bi-directed edge to represent that two variables share some unobserved confounders as input to their structural equations.
>
> In general, our experiments introduce confounding bias in an artificial, fully observed setting, noting that confounding is pervasive in real-world scenarios. Our goal is to close the gap between the assumptions made in RL benchmarks -- such as the absence of confounding – and the conditions encountered in practice.

---

> > ### Comment · Reviewer_E4zm · 2025-08-08
> >
> > Thank you for the response and my questions have been addressed. I have also read other reviewer's comment, and I think this work is a great step towards studing CMDP problems while I look forward to it being extended to a more comprehensive confounding setup (e.g. different parent-children relationships/structure of the confounder towards states and actions). I would like to keep my score as weak accept.

---

> ### Author Response · Authors · 2025-08-06
>
> Dear Reviewer E4zm,
>
> I hope this message finds you well. We would like to kindly follow up on our recent rebuttal and inquire whether you might have had the chance to review our responses. We appreciate the time and effort you are putting into the review process. We believe we have addressed all the concerns raised, but please let us know if anything remains. We are happy to provide further clarifications and to engage in a constructive reviewing process to help us improve the manuscript.
>
> Thank you again for your time and consideration.

---

> ### Comment · Area_Chair_zJdK · 2025-08-06
>
> Please respond to the authors' rebuttal and indicate if you are happy to reconsider your score based on the rebuttal and the other reviewers' comments.

---

### Official Review · Reviewer_eDJW · 2025-07-01

**Clarity:** 1
**Significance:** 3
**Originality:** 2
**Rating:** 4
**Confidence:** 5

**Summary:**

Summary: this paper studies deep RL motivated by RL settings with unobserved confounders. In particular, it focuses on scaling up to DQN the (s,a)-rectangular robust bounds of Zhang and Bareinboim. They convert from the probabilistic form of the bounds to an expectation which can be sampled for DQN updates. The paper develops masked versions of 12 Atari games and trains DQN methods with these lower bounds.

**Questions:**

Questions/suggestions:

I think something that would help the experimental design significantly is 1) compare to methods that explicitly regularize for coverage from the behavior policy, like CQL, or more importantly 2) consider an ablation where the robust bounds from Zhang and Bareinboim are replaced with a uniform lower bound on the reward (i.e, coming from the environment).

I think the more important ablation to run is 2) as it helps isolate whether the PI bounds are being informative in some regions, or whether the method is improving on vanilla DQN because of variance regularization and penalizing policies that stray from the behavior policy; it should also be very runnable given the existing codebase.

**Ethical Concerns:**

["NO or VERY MINOR ethics concerns only"]

**Final Justification:**

The ablations that the authors introduced were helpful. I thought the most useful thing was the set of confounded environments. The method flows naturally from prior work, and so the paper would be improved by better connecting to other offline RL frameworks.

**Quality:**

2

**Strengths And Weaknesses:**

Strengths:
- There's been a lot of recent work lately on RL with unobserved confounders but few works establishing how these methods do in empirical deep RL environments with modern empirical deep RL methods. They find that this can improve upon naive DQN in terms of total reward.
Weaknesses:

significance:

- The paper pitches itself as being close to RL methods for unobserved confounders, but the method itself is much closer to RL methods that constrain to be close to the behavior policy (like CQL). There is not much mention of these methods that should be a more appropriate comparison.

Claims matching up with experimental design: The paper's main novelty is in extending optimizing RL with lower bounds to the realm of deep RL. However, the experimental design can be improved, which would be important as the new method is a fairly direct extension of prior work on partial identification bounds based on the structure of probability distributions alone, without any additional information.


- Are the unobserved confounders chosen here really UCs that affect both state and next action? It's quite confusing from the experimental design whether this is a story of removing extraneous information; or whether the robustness to unobserved confounders is helpful. The scenarios presented in the main text are helpful, but somewhat read as "anecdata". The explanations in the appendix are somewhat confusing, most of them say "The original game screen shows score and remaining life which shouldn’t be the major factor affecting the policies." It sounds like irrelevant information is being removed from the environment, and the criterion for the unobserved confounder selection actually seems to choose parts of the environment that are "action confounders" but not full confounders.

"The core design idea is that we would like to occlude the part of the screens that contains information useful for making decisions but is not a significant factor from human players’ perspectives." - This selection criterion for unobserved confounders seems like it would also omit bona-fide unobserved confounders that affect both action selection by good policies as well as the transition dynamics. For things like score and # lives, conditional on the observed state and action history, they have no further effect on the transition dynamics.

It's OK if this is more of a partial observability or irrelevant information story, but I believe that should be precisely presented (rather than emphasizing unobserved confounders).

---

> ### Author Rebuttal · Authors · 2025-07-31
>
> We thank you for your thoughtful feedback. We appreciate your acknowledgment of the significance of our work and address your concerns in the sequel.
>
> > **W1** “The paper pitches itself as being close to RL methods for unobserved confounders, but the method itself is much closer to RL methods that constrain to be close to the behavior policy (like CQL). There is not much mention of these methods that should be a more appropriate comparison.”
>
> Our method conceptually differs from CQL in that we use pessimism to counter the uncertainty due to unobserved confounders in the offline datasets, while CQL uses pessimism to counter the lack of coverage on the action space, i.e., overly optimistic Q values for actions not visited in the datasets. This means that even when the dataset has full coverage over the action space for each visited state, there could still be unobserved confounders resulting in biased Q-value estimations, which our proposed method can handle.
>
> For concreteness, we provide below a confounded MDP example that differentiates our proposed method from CQL.
>
> **Example**: For simplicity, we consider a single-step CMDP with all variables binary. At each episode, the state $S$ is sampled from a Bernoulli distribution, $P(S=1)=0.5$. The behavioral policy’s action $X$ is decided by $X = S\oplus U_X$, if $U=0$ and $X = \neg S\oplus U_X$, if $U=1$. And the reward $Y$ is decided by $Y = S\oplus X$, if $U=0$ and $Y = \neg S\oplus X$, if $U=1$. $U$ and $U_X$ are unobservable exogenous variables and $U$ is a confounding variable where $P(U=1)=2/3$ and $P(U_X = 1) = q, q \in (0,1)$.
>
> To simplify the discussion, we will focus on the case where state $S=0$ (the following argument can also be easily adapted to the other state). We can calculate the behavioral policy distribution and verify that the behavioral policy has a full coverage over the action space, $\pi_\beta(X=0|S=0) = \frac{1}{3}(1-q) + \frac{2}{3}q > 0$ and $\pi_\beta(X=1|S=0) = \frac{1}{3}q + \frac{2}{3}(1-q) > 0$. For $S=0$, the optimal interventional Q-values are $Q^\*(0, 0) = P(Y=1|S=0, do(X=0)) = \frac{2}{3}$ and $Q^\*(0, 1) = P(Y=1|S=0, do(X=1)) = \frac{1}{3}$. The optimal policy is $do(X=0)$ when $S=0$.
>
> Then, we calculate the CQL Q-values. Given the objective function in equation (4) in the [CQL paper](https://arxiv.org/pdf/2006.04779), we can use Pytorch to calculate the CQL Q-values. When we set $\alpha=1.0$ and $q=1/4$, the learned CQL Q-values at convergence are $Q(0,0) = 0.23 < Q(0,1)=0.27$ suggesting a suboptimal policy. In fact, no matter how we tune the alpha parameter in the CQL objective, there always exists some $U_X$ distributions that can yield a wrong policy under CQL training. Thus, we have demonstrated how CQL can fail under confounding biases.
>
> Instead, we calculate the causal Bellman lower bound to the optimal Q-values. We have $\underline{Q_*}(0, 0) = P(X=0|S=0)P(Y=1|S=0,X=0)=\frac{2}{3}q$ and $\underline{Q_*}(0, 1) = P(X=1|S=0)P(Y=1|S=0,X=1)=\frac{1}{3}q$. Clearly, our causal bound correctly captures the optimal interventional policy $do(X=0)$ when $S=0$.
>
> More broadly, since CQL and our method address different aspects of data bias for off-policy learning, combining those two approaches will yield a more robust offline learning algorithm that can handle both the action space coverage and the unobserved confounding issue.
>
> > **W2** “Are the unobserved confounders chosen here really UCs that affect both state and next action? It's quite confusing from the experimental design whether this is a story of removing extraneous information; or whether the robustness to unobserved confounders is helpful. …It sounds like irrelevant information is being removed from the environment, and the criterion for the unobserved confounder selection actually seems to choose parts of the environment that are "action confounders" but not full confounders. … For things like score and # lives, conditional on the observed state and action history, they have no further effect on the transition dynamics. It's OK if this is more of a partial observability or irrelevant information story, but I believe that should be precisely presented (rather than emphasizing unobserved confounders).”
>
> We would like to reiterate that our unobserved confounders are all chosen from the observation area where the saliency map of the behavioral policy has high values, indicating that the behavioral policy’s action choice is affected by that information. We identified many non-trivial confounders via this approach. For example, in Boxing, we mask out half of the arena and force the agent to stay on the left; in Breakout, we mask out the whole stack of bricks so the agent only learns to shoot the ball back without knowing how many bricks are left; in Choppercommand, we remove the minimap that can reveal incoming enemies out of the main screen; in Gopher, we mask out the gopher location so the agent has to focus on the hole location; in Pong, we mask out opponent’s location; and in Roadrunner, we mask out sky/desert that can indicate levels/incoming obstacles.
>
> Regarding confounders like score and remaining lives, for most of the Atari games we selected (e.g., Kungfu Master, Asterix, Gopher, Pacman), the game difficulty level changes dynamically w.r.t the player’s current score (or remaining time/lives). And conditioning on the limited stacked observations we provide to the agent cannot fully determine the exact level in general. Thus, the observed state transitions are indeed affected by those unobserved confounders. From a human player’s perspective, knowing that having a high score and that the game level is different shouldn’t alter our winning strategy a lot, as the basic game mechanism is still the same, except that it usually requires a faster reaction or more precise execution of the actions. Thus, for DQN agents that can react precisely to the frames (w/o considering sticky actions), we don’t regard this extra information as necessary to learn a good policy. But it doesn’t mean that knowing scores and remaining lives is not helpful. Moreover, given that there could be spurious correlations between remaining time/score/lives and the game mechanism that can be exploited by the DQN, these are genuine confounders.
>
> To summarize, the confounders we chose may either reveal information on the reward/transition functions or directly affect the future game process. Empirically, if those confounders are indeed irrelevant information, we shouldn’t have observed such a drastic performance drop on those standard baseline DQNs with recommended/tuned hyper-parameters.
>
> Having said that, our experiments introduce confounding bias in an artificial, fully observed setting, noting that confounding is pervasive in real-world scenarios. Our goal is not to solve harder Atari variants, but to study and close the gap between the assumptions made in RL benchmarks (such as the absence of confounding) and the conditions encountered in practice.  We appreciate the opportunity to clarify this issue and will also revise the wording in the appendix to reduce confusion.
>
> > **Q1**  “I think something that would help the experimental design significantly is 1) compare to methods that explicitly regularize for coverage from the behavior policy, like CQL, or more importantly 2) consider an ablation where the robust bounds from Zhang and Bareinboim are replaced with a uniform lower bound on the reward (i.e, coming from the environment).”
>
> For Question (1), we have demonstrated in our previous reply to W1 with a detailed example where CQL cannot solve the unobserved confounding case, while the newly proposed Causal Bellman Equation can. More generally, CQL focuses on the challenges of the lack of full action space coverage, called “no overlap” in the causal inference literature, while our work focuses on the challenges due to unobserved confounding. These challenges might seem related, but they model different aspects of data bias in off-policy RL. As we demonstrated in our previous response, confounding bias exists even when the action space has full coverage, and all realizations of actions have been observed.
>
> For Question (2), we train the model with a uniform lower bound while keeping all other hyperparameters the same as the original Causal-DQN. Across all 12 Atari games we tested, the ablation variants all fall far behind Causal-DQN, indicating that an adaptive lower bound is indeed helpful for learning from confounded data.
>
> env/algo|Causal-DQN|UniformBound-DQN
> -|-|-
> Amidar          | 282.6             | 32.8
> Asterix         | 2587.0            | 520.0
> Boxing          | 71.0              | 16.0
> Breakout        | 131.2             | 0.1
> ChopperCommand  | 1658.0            | 1060.0
> Gopher          | 7327.2            | 664.0
> KungFuMaster    | 44196.0           | 14725.0
> MsPacman        | 1747.6            | 486.0
> Pong            | 21.0              | 0.0
> Qbert           | 4458.0            | 262.0
> RoadRunner      | 27414.0           | 3380.0
> Seaquest        | 980.0             | 108.0

---

> > ### Comment · Reviewer_eDJW · 2025-08-06
> >
> > Thanks for the rebuttal response. While I know that the algorithm is conceptually different from CQL, it should be noted that it's much closer to something like CQL than sensitivity analysis that starts from the "no unobserved confounding" assumption that then introduces robustness. Moreover, I think that the findings that UC-robust RL can improve upon non-robust RL are well-explained by the pessimism paradigm, which CQL is representative of, in that robustness can find better policies in offline RL. This is well-studied.
> >
> > Thanks, the ablation with the uniform lower bound are helpful.

---

> ### Author Response · Authors · 2025-08-06
>
> Dear Reviewer eDJW,
>
> I hope this message finds you well. We would like to kindly follow up on our recent rebuttal and inquire whether you might have had the chance to review our responses. We appreciate the time and effort you are putting into the review process. We believe we have addressed all the concerns raised, but please let us know if anything remains. We are happy to provide further clarifications and to engage in a constructive reviewing process to help us improve the manuscript.
>
> Thank you again for your time and consideration.

---

> ### Comment · Area_Chair_zJdK · 2025-08-06
>
> Please respond to the authors' rebuttal and indicate if you are happy to reconsider your score based on the rebuttal and the other reviewers' comments.

---

> ### Author Response · Authors · 2025-08-06
>
> We appreciate the reviewer’s response and are grateful for the opportunity to engage in this discussion.
>
> > Thanks for the rebuttal response. While I know that the algorithm is conceptually different from CQL, it should be noted that it's much closer to something like CQL than sensitivity analysis that starts from the "no unobserved confounding" assumption that then introduces robustness.
>
> First, we would like to respectfully point out that, as demonstrated in the detailed example in our rebuttal (W2), our proposed Causal-DQN addresses a different challenge—unobserved confounding—than CQL, which deals with no overlap. Consequently, our method exhibits distinctly different behaviors in most environments.
>
> Second, we want to clarify that we never claimed our method is based on sensitivity analysis. There are two main approaches for addressing unobserved confounding in non-identifiable effects: (1) bounding the target queries using observed data and (2) sensitivity analysis, which initially imposes additional assumptions to make effects identifiable, then varies the assumptions to see how effect evaluations change. In this work, we adopt the first approach, which bounds the optimal Q-value using confounded observational data (Proposition 3.1).
>
> >Moreover, I think that the findings that UC-robust RL can improve upon non-robust RL are well-explained by the pessimism paradigm, which CQL is representative of, in that robustness can find better policies in offline RL. This is well-studied.
>
> To the best of our knowledge, this work is the first to conduct robust, model-free reinforcement learning with the presence of unobserved confounding in high-dimensional image inputs. **As we have shown in the example, clearly CQL cannot handle UCs** and is not UC-robust at all while our proposed method can correctly learn the optimal policy. If the reviewer is aware of any existing work (other than CQL) that addresses the challenge of unobserved confounding under the same problem setting—without additional parametric knowledge of the latent confounder—we would appreciate it if you could share those references. We would be happy to review them and include them in the updated manuscript.

---

### Official Review · Reviewer_UUDT · 2025-07-01

**Clarity:** 3
**Significance:** 2
**Originality:** 3
**Rating:** 4
**Confidence:** 3

**Summary:**

This paper proposes Causal DQN, a novel deep reinforcement learning algorithm designed to handle unmeasured confounding in offline, high-dimensional environments. Unlike standard DQN methods that assume the absence of hidden confounders, Causal DQN adopts a partial identification perspective and seeks policies that perform well under the worst-case confounding compatible with the data. The method is empirically validated on twelve confounded Atari games, where it consistently outperforms standard DQN baselines in settings with mismatched behavior and target policies due to hidden biases in visual inputs.

**Questions:**

Q1: How sensitive is the proposed Causal-DQN to misspecification of the assumed causal graph (DAG), particularly in cases where the unobserved confounder does not simultaneously affect both the action and the reward/transition as modeled in the CMDP framework? Moreover, while the experiments are conducted on Atari games, is there empirical justification or diagnostic evidence that the confounding structures in these games align with the assumed DAG used in the method?

Q2: Is the proposed method limited to environments with discrete action spaces and image-based observations? All experiments are conducted on Atari games with discrete actions and visual inputs.

Q3: The confounding in the experiments is induced via deliberate masking of visual features, which aligns well with the assumed causal graph. However, in real-world scenarios, unobserved confounders may arise from unknown or unrecorded variables rather than masked inputs. Could the authors clarify whether and how the proposed Causal-DQN is expected to mitigate the influence of more general or latent confounding, especially when the nature of confounding is less structured or directly observable?

Q4: If I understand correctly, the causal Bellman lower bound used in this work is based on the theoretical framework proposed by Zhang & Bareinboim (2025), and this paper is the first to integrate it into a practical DQN algorithm with empirical validation in high-dimensional visual environments like Atari. Is this a fair characterization of the novelty and contribution of the work? -> I would say the paper has somewhat weaker motivation and is borderline for NeurIPS.

**Ethical Concerns:**

["NO or VERY MINOR ethics concerns only"]

**Limitations:**

Yes

**Quality:**

3

**Strengths And Weaknesses:**

S1: This paper addresses a novel and important problem in deep reinforcement learning: learning from off-policy data in the presence of unobserved confounders.
S2: The paper is well-structured.

For weakness, please see my questions.

---

> ### Author Rebuttal · Authors · 2025-07-31
>
> We are grateful for your detailed comments and for acknowledging our work’s importance and novelty. We have addressed your questions in the sequel.
>
> > **Q1** “How sensitive is the proposed Causal-DQN to misspecification of the assumed causal graph (DAG), particularly in cases where the unobserved confounder does not simultaneously affect both the action and the reward/transition as modeled in the CMDP framework? Moreover, while the experiments are conducted on Atari games, is there empirical justification or diagnostic evidence that the confounding structures in these games align with the assumed DAG used in the method?”
>
> Yes, our method is robust to the possible model misspecification when the unobserved confounder does not simultaneously affect both the action and the reward/transition. Our proposed algorithm is still able to obtain valid bounds over the optimal value function in such settings. Indeed, CMDP is a general class of models that subsumes standard MDP. Therefore, it can handle different confounding situations between those variables. As we don’t restrict how the confounder is used inside transition functions/reward functions, CMDP can handle the situations where those functions are actually not confounded, and the confounder in the function signature serves only as a dummy input.
>
> Also, the confounders used in our Atari experiments, as entailed in the experiment section, are generated via identifying the part of the screen that the behavioral agent attends to and then masking out a part such that the behavioral agent’s performance would drop. For example, one could verify the significance of masked features (to the behavioral policy) in the saliency map in Figure 2(a). This procedure ensures that the behavioral agent is indeed using those masked areas to make decisions, and they have a meaningful correlation with the state transitions and rewards.
> More broadly, this is our approach to introducing confounding in an artificial, fully observed setting, acknowledging that confounding is pervasive in real-world scenarios. Our goal is to close the gap between the assumptions made in RL benchmarks -- such as the absence of confounding – and the conditions encountered in practice.
>
> > **Q2** ”Is the proposed method limited to environments with discrete action spaces and image-based observations? All experiments are conducted on Atari games with discrete actions and visual inputs.”
>
> Similar to the standard DQN, our proposed method handles settings with discrete actions and a continuous state space. Given the popularity of DQN and the fact that it serves as the foundation for a large class of value-based deep RL algorithms, we believe our proposed algorithm covers a wide range of problem cases. That said, to the best of our knowledge, this is the first paper to address such considerations in these settings, and we hope it can initiate a conversation within the community, one that we can build on to expand the general toolbox for more complex decision-making problems.
>
> > **Q3** “The confounding in the experiments is induced via deliberate masking of visual features, which aligns well with the assumed causal graph. However, in real-world scenarios, unobserved confounders may arise from unknown or unrecorded variables rather than masked inputs. Could the authors clarify whether and how the proposed Causal-DQN is expected to mitigate the influence of more general or latent confounding, especially when the nature of confounding is less structured or directly observable?”
>
> Yes, our proposed method is able to handle the general form of unobserved confounding since our CMDP model encodes the least amount of assumptions for representing the presence of unobserved confounding in the MDP environment. Specifically, we model unobserved confounding as an exogenous variable $U$ that could be further decomposed into multiple smaller variables and has a rich structure. However, we do not intend to impose any additional structural knowledge in CMDPs, which allows our proposed Causal-DQN to be applicable to any MDP environment where the confounding bias could not be excluded a priori. Also, we want to clarify that masking is a straightforward way to simulate unrecorded variables. For example, in the pong game, we completely mask out the opponent’s location $U_t$, making such information totally unobserved to the learner.
>
>
> > **Q4** “If I understand correctly, the causal Bellman lower bound used in this work is based on the theoretical framework proposed by Zhang & Bareinboim (2025), and this paper is the first to integrate it into a practical DQN algorithm with empirical validation in high-dimensional visual environments like Atari. Is this a fair characterization of the novelty and contribution of the work? -> I would say the paper has somewhat weaker motivation and is borderline for NeurIPS.”
>
> Theoretically, we propose the first confounding-robust off-policy learning algorithm that bounds the target Q-value using function approximation. The derivation of the learning updates and how we combine them into a sample-based learning method like DQN is non-trivial. Also, Zhang & Bareinboim (2025) only consider the policy evaluation problem, and do not provide policy recommendations.  In this work, we study the more challenging policy learning setting, and Causal-DQN is able to obtain an effective conservative policy from confounded observations.
>
> Empirically, to the best of our knowledge, we are the first to perform confounding robust learning methods for high-dimensional environments (like Atari games) with function approximations, while the previous paper like (Zhang & Bareinboim 2025) only handles small discrete state spaces like very small grid worlds.

---

> > ### Comment · Reviewer_UUDT · 2025-08-05
> >
> > Thank you to the authors for their detailed and thoughtful response. While some concerns remain regarding the realism of the confounding setup and broader applicability, I appreciate the clarification of the CMDP’s flexibility and the theoretical and empirical contributions beyond prior work. The integration of causal bounds into DQN with empirical validation in high-dimensional settings is a meaningful step. I will maintain my borderline accept score.

---

> > > ### Author Response · Authors · 2025-08-07
> > >
> > > Thank you for your engagement in the discussion. We believe we have provided sufficient information on the confounding setup and broader applicability of our proposed method given the pervasiveness of confounding biases in real world decision making problems. Could you elaborate more on your remaining concerns? We are happy to explain in more details.

---

### Official Review · Reviewer_czit · 2025-07-03

**Clarity:** 2
**Significance:** 3
**Originality:** 3
**Rating:** 4
**Confidence:** 4

**Summary:**

This paper proposed a DQN algorithm for an off-line RL environment in the presence of an unobserved confounder. The experiment and explanations in the paper primarily address the source of confounding as missing state information during the data collection stage by behavioral policy. This paper also mentions earlier works such as CMDP and the closed-form expression of the Q-function in the presence of an unobserved confounder. The contribution is to extend the optimal value function with a lower bound and propose an off-line learning algorithm. In practice, it can be very challenging to engineer an RL environment with unobserved confounders. This paper shows 12 modified Atari domains based on salience map analysis.

**Questions:**

Please confirm if the scope of the paper is correct. The decision problem in this paper addresses the off-line RL problem, where an agent has sampled the experience before. The unobserved confounder enters the problem by assuming that the agent that generated the experience might have observability on the state variables that are not present in the collected dataset (Line 60-62).
To me, defining CMDP seems like it will address the scenario beyond what's described above.

Figure 1 illustrates that the observed action, the next state variable, the observed action and reward, and finally the reward and the next state variable can be confounded. In Line 77, when the data generation process is explained, it appears that only one unobserved confounder variable ($U_t$) is considered. Do you also cover the factored state and action space MDP?

In the appendix, is there a proof of Eq. (3) on the Bellman equation for the given setting, or is it from some earlier works?

Figure 2 and Example 1 are too extreme. It removes the opponent in the game and trains a conventional DQN agent that needs to learn a policy by observing the ball only. I don't think this is a realistic example of an unobserved confounder, although it can be used to contrast the performance as shown in Figure 2(d). In this case, how does employing a behavior policy perform in the modified state space?

The notations in Proposition 3.1 require further elaboration, particularly those with underscores.
Does the lower bound hold whether the value function is identifiable or not?


In Algorithm 1, it is an offline method that stores the entire dataset in memory. How to sample transitions in an offline environment? Does the step that samples the initial state $s_1$ work? How can the behavior policy sample an action with a modified state space if it were trained in the original state space? What is the added complexity of the additional sampling in Line 196? In line 204, the opponent is unobserved to the offline learning agent. Does it also see the opponent's location?


In line 243, the optimal policy stipulates that it only needs to see the ball location and its paddle, which is the off-line setting that removes the opponent's visual information and the scoreboard. Then, why did the traditional DQN fail?

Besides, the evaluations were done over 10 episodes across 5 seeds in Figure 8. Would it be possible to enhance the traditional DQN by utilizing different hyperparameters or exploration strategies? I think the evaluation may be favorable to the proposed method.
Following the explanation, the proposed method operates by sampling the transition using both the behavioral policy and the reward. While learning the Q-network, it samples additional points and takes the pessimistic value instead of committing to a single state in Eq. (11).

**Ethical Concerns:**

["NO or VERY MINOR ethics concerns only"]

**Limitations:**

YES

**Quality:**

2

**Strengths And Weaknesses:**

**Strength**
The strength of this paper is to handle the ATARI domain in a causal RL setting, which is a non-trivial contribution.
Many existing benchmarks and algorithms assume the presence of causal variables or causal graphs.
The experiment is very novel that I haven't seen in other papers yet.

**Weakness**
The role of the unobserved confounder in the problem needs to be discussed in depth. The data generation process doesn't fit well with the data corruption process used in the experiment.
From the RL evaluation viewpoint, the experiment itself is insufficient in terms of the number of trials and hyperparameter optimization.
There are supporting theoretical results, such as Proposition 3.1 and the equations in the paper, that need to be elaborated upon in the paper.

---

> ### Author Rebuttal · Authors · 2025-07-31
>
> We thank you for your thoughtful feedback. We appreciate your acknowledgment of the significance of our work and have addressed your concerns in the sequel.
> > **W1** “The role of the unobserved confounder in the problem needs to be discussed in depth. The data generation process doesn't fit well with the data corruption process used in the experiment. From the RL evaluation viewpoint, the experiment itself is insufficient in terms of the number of trials and hyperparameter optimization. There are supporting theoretical results, such as Proposition 3.1 and the equations in the paper, that need to be elaborated upon in the paper.”
>
> On the role of the unobserved confounders and also how the data generation process (definition) aligns with our actual experiments, we will discuss in detail in our replies to **Q1** and **Q2**. Briefly, those unobserved confounders represent information that cannot be observed by the student agent but has been used during off-policy data collection by the behavioral policy. In our experiment, those masked out pixel features together compose a high-dimensional hidden confounder.
>
> Regarding the experiment evaluation, we conducted hyper parameter tuning on a subset of the training parameters and followed the best practice for others. We also tried different DQN variants. For each experiment, we conducted five random trials and reported the normalized mean/IQM/median. We will further explain our evaluation design in reply to **Q8**.
>
> For elaborations on Proposition 3.1 and related equations, we will provide more details in reply to **Q3** and **Q5**. To summarize, those underscored notations represent their counterparts under the worst case environment that could happen given behavioral trajectories. Proposition 3.1 provides a Bellman equation-like update rule to calculate the pessimistic Q-values with guaranteed convergence.
>
> > **Q1** “Please confirm if the scope of the paper is correct. … defining CMDP seems like it will address the scenario beyond what's described above.”
>
> We consider a similar off-policy learning setting as the original DQN paper, except that the sample trajectories are not generated by the student DQN being trained, but from a more complex teacher model with better sensor capabilities. Those state variables that cannot be observed by the student agent become unobserved confounders $U_t$ in the CMDP, affecting the teacher model’s action selection, next state transitions, and the rewards. For example, in the Pong game, we masked out the opponent’s paddle location, which will surely affect those factors mentioned above. Thus, we find that defining CMDP is necessary and adequate for our claimed scope of the paper.
>
> > **Q2** “In Line 77, when the data generation process is explained, it appears that only one unobserved confounder variable ($U_t$) is considered. Do you also cover the factored state and action space MDP?”
>
> Yes, our framework also applies to factored state/action space MDPs. The confounding variable U can be high-dimensional, supporting the situation where multiple variables in the environment are unobserved by the student agent and thus become unobserved confounders.
>
> > **Q3** “In the appendix, is there a proof of Eq. (3) on the Bellman equation for the given setting, or is it from some earlier works?”
>
> When the experimental data is available and the learner can collect data by directly intervening in the environment, the CMDP is reduced to a standard MDP, and the Bellman equation of Eq. (3) follows from the standard RL literature. A detailed survey can be found in [6].
>
> > **Q4** “Figure 2 and Example 1 are too extreme. ... I don't think this is a realistic example of an unobserved confounder, although it can be used to contrast the performance as shown in Figure 2(d). In this case, how does employing a behavior policy perform in the modified state space?”
>
> First, we respectfully disagree that the given situation is not realistic, as there are many examples in practice where the agent’s sensors have been damaged, and it cannot perceive information far from its position (e.g., the opponent's location). In this case, one could have two choices: (1) stop the agent entirely, or (2) realign the agent’s current policy to a more conservative, yet still effective one. In the Pong game setting, while the agent could only track the ball location, there exists a winning strategy in the confounded Pong game: one only needs to shoot the ball back every time and wait for the opponent to make a mistake. The strategy itself is even more straightforward than directing the ball smartly to a direction that is further away from the opponent’s current position. When we deploy the behavioral policy in the confounded Pong game, the agent can achieve a score of around 18, which is still less than the optimal score of 21 achieved by a teacher agent who could observe the full state. This means that while the conservative policy of ball tracking is not truly optimal (at least not as in the fully observed case), it can still be effective in practice and is learnable from confounded observations.
>
> > **Q5** “The notations in Proposition 3.1 require further elaboration, particularly those with underscores. Does the lower bound hold whether the value function is identifiable or not?”
>
> Yes, the bounds in Prop 3.1 hold for any CMDP, regardless of whether the target is identifiable or not. Specifically, when the value function is not identifiable, our proposed lower bound is tight (unless further assumptions are provided). When the value function is identifiable, our proposed lower bound remains valid because it compensates for the Bellman updates with the worst-case scenario, ensuring that any real transitions under the identifiable case are at least as good as this one. So it’s still a valid lower bound.
>
> > **Q6** “How to sample transitions in an offline environment? Does the step that samples the initial state work? How can the behavior policy sample an action with a modified state space if it were trained in the original state space? What is the added complexity of the additional sampling in Line 196? In line 204, the opponent is unobserved to the offline learning agent. Does it also see the opponent's location?”
>
> The environment is a standard off-policy environment. The learning agent passively observes the behavioral agent interacting with the unmodified environment, collects the observed trajectories (masked, not including the unobserved confounders), and then saves them into the replay buffer. Those masked trajectories are then used to train our DQN policy. As for Line 196, in implementation, we simply use the observed worst state values during training from the replay buffer as an empirical estimation. So, there is no sampling overhead in practice. In line 204, our causal DQN policy cannot see the opponent’s location because that’s part of the masked area; it only makes a decision based on the observed ball location.
>
> > **Q7** “In line 243, the optimal policy stipulates that it only needs to see the ball location and its paddle, which is the off-line setting that removes the opponent's visual information and the scoreboard. Then, why did the traditional DQN fail?”
>
> Traditional DQN fails to learn since it does not account for the confounding bias. Such bias could make some observed actions appear more effective than they actually are. For example, in the teacher’s trajectories, the behavioral policy only places the ball in the middle when the opponent is located at either corner (top or bottom), and is unable to cover the middle. Therefore, when not observing the opponent’s location, it would appear that placing the ball in the middle is always followed by a positive score, making it a winning strategy. However, if the traditional DQN picks such a policy only based on the observed value function, not accounting for the confounding bias, it would lead to sub-optimal performance, since the opponent could return the ball in the middle in most situations (except when it is at the corners). On the other hand, using the newly proposed causal bound, the agent lowers the weight on the observed behavioral trajectories by its empirical policy distributions. This allows the Causal-DQN agent to learn conservative Q-values without falling into such overly optimistic Q-value traps.
>
> > **Q8** “Would it be possible to enhance the traditional DQN by utilizing different hyperparameters or exploration strategies? I think the evaluation may be favorable to the proposed method.”
>
> To the best of our knowledge, it is unlikely due to confounding bias. Specifically, we tuned a subset of training hyperparameters for all DQN variants, like learning rates, batch sizes, and replay buffer sizes (lines 220-226). We keep the comparison fair by using the same set of hyper-parameters for all DQN variants and following the best practices as entailed in the original DQN paper.
>
> We also explored multiple DQN variants to see if adding more data or adding LSTM layers can help mitigate the learning difficulties. In lines 210-219, we described three different DQN baselines, including: (1) CNN-DQN with trajectories from behavioral policy but observations are masked; (2) LSTM-DQN with the same type of trajectories as (1); and (3) CNN-DQN with masked trajectories collected by itself (the same setting as the standard DQN). However, from Table 1, we see that none of these variants can learn an effective policy in the confounded environments.
>
> By and large, we believe the newly proposed, confounded-robust method is significant, since in most practical settings where RL may be used, unobserved confounders and the corresponding trajectories are the rule rather than the exception. There is a natural mismatch between the perceptual capabilities of most agents and the assumptions made in much of the current literature, which either implicitly assumes a perfect match or derives correctness results under that assumption.

---

> > ### Comment · Reviewer_czit · 2025-08-06
> >
> > **W1:**
> > The weakness still remains.
> > Rebuttal didn't clearly justify or elaborate on the role of UC.
> > It mentioned that it is aligned and the following Q1 and Q2 don't provide more information or discussion. The rebuttal focused on defending the question but those questions are requesting confirmation or more details.
> >
> > 5 trials in RL experiment sounds like too small, in general sense.
> > It is not clear how much hyper-parameter tuning has done. It seems that the experiment took the best known working combination and applied it to all uniformly.
> >
> >
> > **Q1:**
> > Here, the student model is the DQN that is only trained in an offline manner.
> > Therefore, it cannot generate the full trajectory including the removed random variables.
> > My question was that CMDP could cover scenarios other than the one described in this paper.
> >
> > **Q2:**
> > If the proposed method can handle factored state/action space,
> > all the equations (1), (2), (3), (4), ... needs to be checked.
> > Does everything hold by replacing s_t with a vector of random variables?
> >
> > **Q4:**
> > My question was that example 1 and (other problems) are created in an adversarial way by removing essential information for solving the problem, and I think it is too extreme.
> >
> >
> > **Q5:**
> > In the response, it is said that,
> > The lower bound is tight when value is not identifiable/The lowe bound is valid when value is identifiable.
> >
> > Is it correct?
> >
> >
> > **Q6:**
> > When the full trajectories are available in offline training, why should we mask important state variables? This might be justified by stating that we want to train another agent that cannot have access to the masked state variables.
> > Is masking an essential step for generating samples for training the student agent?
> >
> > **Q7:**
> > This question was asked because of the following sentence in line 243.
> > "Thus, an intuitive optimal policy should only look at the ball location and the agent’s own paddle location to decide the move."
> >
> > Then, can we say that the actually good policy is counter-intuitive due to the confounder?
> >
> > **Q8:**
> > Update freq fixed to 20, Batch size 512, Buffer 100k, learning rate 5e-4.
> > Other hyperparameters are from [38; the nature DQN paper].
> >
> > This reads to me as if no hyperparameter tuning has been done for each setting, but applying a common set.

---

> > > ### Author Response · Authors · 2025-08-08
> > >
> > > Thank you for your engagement in the discussion.  We will reply to your concerns in the following.
> > >
> > > > **W1**: The weakness still remains. Rebuttal didn't clearly justify or elaborate on the role of UC. It mentioned that it is aligned and the following Q1 and Q2 don't provide more information or discussion. The rebuttal focused on defending the question but those questions are requesting confirmation or more details.
> > >
> > > First, we would like to clarify that we have attempted to be detailed in our response, as we believe there are some misunderstandings regarding our problem settings and the concept of confounding bias in general. We have attempted to provide background information and detailed examples to help clarify the issue.
> > >
> > > Regarding the role of unobserved confounding, it is prevalent in many real-world applications and has been a primary cause of distribution shifts between observational and experimental data. For example, healthcare records data suggested Remdesivir was effective for Covid-19 infections, and even received a temporary FDA approval for medical use. However, later randomized trials revealed it was not effective, and its observed effectiveness was due to unobserved confounding factors (e.g., income levels) that result in physicians administering exploratory drugs to patients with access to better healthcare. Several robust policy learning methods have been proposed to address confounding bias in causal inference. However, these methods are often limited to discrete or low-dimensional domains. In this work, we attempt to address this issue by combining the computational framework of DQN with the partial identification methods in causal inference. Our example focuses on modified Atari games since we believe they capture the unique challenges of performing robust causal inference in high-dimensional domains.
> > >
> > > In CMDPs, the unobserved confounders are the latent state factors that can potentially affect the next state transitions, actions selected by the behavioral policy, and the reward functions. These confounders are not recorded in the observed trajectories. In the healthcare domains, they are the latent factors that introduce spurious correlations between drug assignment and health outcomes. In the Chopper Command game, they are the minimap (radar sensors) that show the position of future enemies. If the reviewer has any specific questions about this definition of unobserved confounders and their impacts, we would be more than happy to answer them.
> > >
> > > > **W1**: …. 5 trials in RL experiment sounds like too small, in general sense. It is not clear how much hyper-parameter tuning has done. It seems that the experiment took the best known working combination and applied it to all uniformly.
> > >
> > > Regarding the number of trials, we believe 5 trials are a reasonable amount to draw meaningful conclusions. Note that in the literature, the original PPO paper used 3 trials, SAC paper used 5 trials and the teacher model we used from a 2024 paper also used 5 trials. Though we agree that using more random trials may derive more statistically robust conclusions, this won’t change the overall direction of the conclusion that our proposed Causal-DQN can handle UCs in high-dimensional settings while standard off-policy learners cannot.
> > >
> > > > **Q1**: Here, the student model is the DQN that is only trained in an offline manner. Therefore, it cannot generate the full trajectory including the removed random variables. My question was that CMDP could cover scenarios other than the one described in this paper.
> > >
> > > First, as mentioned earlier, CMDPs could model the challenges of unobserved confounders in the sequential decision-making settings where the Markov property holds. Besides examples in this paper, it also includes designing dynamic treatment regimes from healthcare records and evaluating the effects of social programs. In this paper, we focus on the off-policy learning settings where the data is generated by a different behavioral policy [48].
> > >
> > > Also, we are a bit confused about what the reviewer means by “it cannot generate the full trajectory including the removed random variables.” DQN and proposed causal-DQN are model-free algorithms that do not learn the underlying system dynamics. So, it cannot generate trajectories. These algorithms will learn a policy mapping that looks at the observed states (after masking) and proposes an action. Could the reviewer further elaborate on this question?

---

> > > > ### Author Response · Authors · 2025-08-08
> > > >
> > > > > **Q2**: If the proposed method can handle factored state/action space, all the equations (1), (2), (3), (4), ... needs to be checked. Does everything hold by replacing s_t with a vector of random variables?
> > > >
> > > > The derivations of Eqs. (1-4) have been well-discussed in the literature on dynamic treatment regimes, e.g., Murphy (2005) and causality textbooks. We only stated them here because we don’t want to overstate the results. The derived model imposes no constraints on the domains of variables and still holds when states are replaced with vector variables. Still, for the sake of completeness, we provide a brief justification of them below.
> > > >
> > > > Equation 1: For the clarity of the derivations, we can write the transition distribution $\mathcal{T}$ and reward functions $\mathcal{R}$ as $P(s_{t+1}|s_t, x_t)$ and $P(y_t|x_t, s_t)$, respectively. Then, following the basic conditional probability rules, we can decompose the joint distribution by the topological ordering of the variables as $P_\pi = P(s_1)P(x_1|s_1)P(y_1|s_1, x_1)...P(y_T|s_1, …, s_T, x_1, …, x_T, y_1, …, y_{T-1})$. And then by the Markov condition encoded by the CMDP definition, we can remove all other variables in the conditioning set other than the variables in the same time step. This will lead us to equation 1.
> > > > Equation 2: First, given the current state $S_t$, all backdoor paths from each action $X_t$ to future states and rewards are blocked. Thus, by rule 2 of do-calculus (Pearl 2000, Chapter 3.4), it’s equivalent to setting actions with atomic interventions $do(x_t)$ where $x_t \sim \pi(x_t|s_t)$. Then by the interventional distribution definition of structural causal models from which the CMDP definition is derived (equation 7.2 in [Pearl’s Causality textbook](https://bayes.cs.ucla.edu/BOOK-2K/)), we have equation 2.
> > > > Equation 3: this follows the standard Bellman Optimality Equation.
> > > > Equation 4: From CMDP definition, we can see that state $S_{t}$ blocks all backdoor path from $X_t$ to $S_{t+1}$, i.e., paths starting with an arrow pointing into $X_t$. Thus, by do-calculus rule 2 (Pearl 2000, Chapter 3.4), we can identify the true environment transition distribution by the observational data, $P(s_{t+1}|s_t, do(x_t)) = P(s_{t+1}|s_t, x_t)$. The same procedure also applies to rewards.
> > > >
> > > > > **Q4**: My question was that example 1 and (other problems) are created in an adversarial way by removing essential information for solving the problem, and I think it is too extreme.
> > > >
> > > > First, we do not consider our examples to be extreme, as there is still enough information available to effectively play the game, as demonstrated in our experiments. Nonetheless, this discussion seems somewhat subjective and ultimately comes down to personal preference.
> > > >
> > > > A more important question here, however, is whether we should study policy learning even when the environment is adversarial and imperfect. We believe the answer is yes, as addressing adversarial settings has been a fundamental motivation for safe and robust learning. These scenarios often reflect real challenges in many practical domains, and the ability to navigate these adversarial situations is indicative of strong artificial intelligence.
> > > >
> > > > For instance, in the Chopper Command game, the minimap acts like a radar sensor that could often be compromised on the battlefield, introducing confounding bias in the data. In this situation, one might either (1) discard all previous data and give up, or (2) continue to adjust the policy and find a way to survive. In this work, we take the second approach and investigate policy learning in high-dimensional space with confounding bias.
> > > >
> > > > > **Q5**: In the response, it is said that, The lower bound is tight when value is not identifiable/The lowe bound is valid when value is identifiable. Is it correct?
> > > >
> > > > Yes, this is correct.
> > > >
> > > > > **Q6**: When the full trajectories are available in offline training, why should we mask important state variables? This might be justified by stating that we want to train another agent that cannot have access to the masked state variables. Is masking an essential step for generating samples for training the student agent?
> > > >
> > > > We are unsure of the meaning of “Is masking an essential step for generating samples for training the student agent?” Masking is essential to **simulate** the presence of unobserved confounders in the experiments, as we want to create latent essential states that will affect both the observed actions and subsequent outcomes. These masked samples are then fed to the student agent for training. However, we want to clarify that masking is only a technique to simulate confounding bias. It is unrelated to the proposed algorithm and its performance.

---

> > > > > ### Author Response · Authors · 2025-08-08
> > > > >
> > > > > > **Q7**: This question was asked because of the following sentence in line 243. "Thus, an intuitive optimal policy should only look at the ball location and the agent’s own paddle location to decide the move." Then, can we say that the actually good policy is counter-intuitive due to the confounder?
> > > > >
> > > > > Yes, when there are confounders, a good policy in this case is different from the one with full observability. More specifically, under UCs, a good policy is usually a conservative one because it lacks the definite information on the environment transitions or rewards.
> > > > >
> > > > > > **Q8**: Update freq fixed to 20, Batch size 512, Buffer 100k, learning rate 5e-4. Other hyperparameters are from [38; the nature DQN paper]. This reads to me as if no hyperparameter tuning has been done for each setting, but applying a common set.
> > > > >
> > > > > Thanks for pointing this out. We actually did an informal hyper parameter search on both confounded Pong and Boxing games. Though due to the high computational cost, we didn’t run a full grid search over all the hyper parameters. But the update frequency, batch size, learning rate and buffer size are roughly the best combination selected from the space of (5, 10, 20) x (128, 256, 512) x (1e-5, 2.5e-4, 1e-4) x (10K, 50K, 100K). We will revise the experiment section to reflect this.

---

> ### Author Response · Authors · 2025-08-06
>
> Dear Reviewer czit,
>
> I hope this message finds you well. We would like to kindly follow up on our recent rebuttal and inquire whether you might have had the chance to review our responses. We appreciate the time and effort you are putting into the review process. We believe we have addressed all the concerns raised, but please let us know if anything remains. We are happy to provide further clarifications and to engage in a constructive reviewing process to help us improve the manuscript.
>
> Thank you again for your time and consideration.

---

> ### Comment · Area_Chair_zJdK · 2025-08-06
>
> Please respond to the authors' rebuttal and indicate if you are happy to reconsider your score based on the rebuttal and the other reviewers' comments.

---

### Note · Authors · 2025-08-14

Dear Reviewers, AC, SAC,

We sincerely appreciate your continued engagement over the past week. This is a valuable opportunity for us to highlight the key contributions of our paper and to address some of the misunderstandings reflected in the reviews. Your feedback is important to us, and we are thankful for the chance to clarify these points.

**Theoretical contribution**: We propose, to our knowledge, the first confounding-robust off-policy learning algorithm that bounds the target Q-value under function approximation. The derivation of the learning updates and their integration into a sample-based method like DQN is non-trivial and technically novel.

**Empirical Contribution**: We are the first to apply confounding-robust learning methods to high-dimensional environments, such as Atari games, with function approximation. Our experiments introduce reasonable confounding biases in an artificial but fully observed setting, illustrating their pervasive nature in real-world scenarios. Our aim is not only to achieve high scores on standard Atari benchmarks, but also to bridge the gap between reinforcement learning benchmark assumptions (i.e., the absence of confounding) and the realities of practical deployment.

**Key clarification**: Our method specifically addresses the bias arising from confounded offline data, not the Q-value overestimation resulting from the narrow state-action space coverage, which CQL targets. As demonstrated in our rebuttal to Reviewer eDJW, even when the action space is visited sufficiently in the offline dataset, unobserved confounding can still cause CQL to learn an incorrect policy, whereas our method correctly solves the problem.

Finally, we would like to express our gratitude for your insightful feedback, which we will incorporate into our updated manuscript. Addressing confounding bias is a widespread challenge across many disciplines. Existing methods in the causal inference literature are typically limited to discrete or low-dimensional domains. This work, for the first time, combines the principles of causal inference with the computational framework of deep reinforcement learning. We believe this presents a significant breakthrough toward confounding-robust policy learning in complex, high-dimensional domains.

Thank you for your continued attention and engagement.

Sincerely,
Authors of Paper #18850

---

### Decision · Program_Chairs · 2025-09-17

**Decision:**

Accept (poster)

**Comment:**

Summary of the paper:

The paper proposes a DQN algorithm for an off-line RL environment in the presence of an unobserved confounder. The experiment and explanations in the paper primarily address the source of confounding as missing state information during the data collection stage by the behavioral policy. The contribution is to extend the optimal value function with a lower bound and propose an off-line learning algorithm. In practice, it can be very challenging to engineer an RL environment with unobserved confounders. This paper shows 12 modified Atari domains based on salience map analysis.

Summary of the discussion:

All reviewers agree on acceptance, but some have still some concerns.

One reviewer had some minor comments:

(1) CARL with UC can address more general scenarios, and either authors could sketch some more in the paper (intro/background/future work) or clearly state the boundary of the current work

(2) Some derivations are said to be trivial, but it is always good to add the details. For example, the recursive form of the Bellman equation can be derived from the basic definition of the value. Factored state extension is not as trivial as the authors replied.

(3) The experiment is not of high quality. Authors mentioned that they saw some papers reporting three trials or five trials, but they are some exceptions. In addition, the experiment is not fully persuasive to draw a conclusion. It might be possible that some better hyperparameters would lead to better results than the complete failures that we see for DQNs.

One reviewer still has some concerns remain regarding the realism of the confounding setup and broader applicability.

Recommendation:

All reviewers vote for acceptance. I, therefore, recommend acceptance and encourage the authors to use the feedback provided to improve the paper for its final version, especially, to address the remaining concerns of the reviewers.